# C-Learning: Horizon-Aware Cumulative Accessibility Estimation

**Panteha Naderian, Gabriel Loaiza-Ganem, Harry J. Braviner, Anthony L. Caterini, Jesse C. Cresswell & Tong Li**
Layer 6 AI
`{panteha, gabriel, harry, anthony, jesse, tong}@layer6.ai`

**Animesh Garg**
University of Toronto, Vector Institute, Nvidia
`garg@cs.toronto.edu`

## Abstract

Multi-goal reaching is an important problem in reinforcement learning needed to achieve algorithmic generalization. Despite recent advances in this field, current algorithms suffer from three major challenges: high sample complexity, learning only a single way of reaching the goals, and difficulties in solving complex motion planning tasks. In order to address these limitations, we introduce the concept of *cumulative accessibility functions*, which measure the reachability of a goal from a given state within a specified horizon. We show that these functions obey a recurrence relation, which enables learning from offline interactions. We also prove that optimal cumulative accessibility functions are monotonic in the planning horizon. Additionally, our method can trade off speed and reliability in goal-reaching by suggesting multiple paths to a single goal depending on the provided horizon. We evaluate our approach on a set of multi-goal discrete and continuous control tasks. We show that our method outperforms state-of-the-art goal-reaching algorithms in success rate, sample complexity, and path optimality. Our code is available at `https://github.com/layer6ai-labs/CAE`, and additional visualizations can be found at `https://sites.google.com/view/learning-cae/`.

## 1 Introduction

Multi-goal reinforcement learning tackles the challenging problem of reaching multiple goals, and as a result, is an ideal framework for real-world agents that solve a diverse set of tasks. Despite progress in this field (Kaelbling, 1993; Schaul et al., 2015; Andrychowicz et al., 2017; Ghosh et al., 2019), current algorithms suffer from a set of limitations: an inability to find multiple paths to a goal, high sample complexity, and poor results in complex motion planning tasks. In this paper we propose $C$-learning, a method which addresses all of these shortcomings.

Many multi-goal reinforcement learning algorithms are limited by learning only a single policy $\pi(a|s, g)$ over actions $a$ to reach goal $g$ from state $s$. There is an unexplored trade-off between reaching the goal reliably and reaching it quickly. We illustrate this shortcoming in Figure 1a, which represents an environment where an agent must reach a goal on the opposite side of some predator. Shorter paths can reach the goal faster at the cost of a higher probability of being eaten. Existing algorithms do not allow a dynamic choice of whether to act safely or quickly at test time.

The second limitation is sample complexity. Despite significant improvements (Andrychowicz et al., 2017; Ghosh et al., 2019), multi-goal reaching still requires a very large amount of environment interactions for effective learning. We argue that the optimal $Q$-function must be learned to high accuracy for the agent to achieve reasonable performance, and this leads to sample inefficiency. The same drawback of optimal $Q$-functions often causes agents to learn sub-optimal ways of reaching the intended goal. This issue is particularly true for motion planning tasks (Qureshi et al., 2020), where current algorithms struggle.

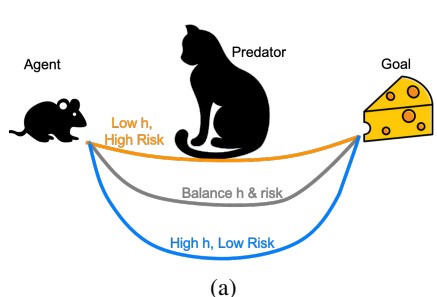 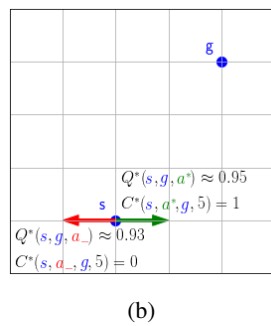

(a)  (b)

Figure 1: $(a)$ A continuous spectrum of paths allow the mouse to reach its goal faster, at an increased risk of disturbing the cat and being eaten. $(b)$ $Q^*$ (with $\gamma = 0.99$) needs to be learned more accurately than $C^*$ to act optimally. The goal $g$ can be reached in $h^* = 5$ steps from $s$, so that $Q^*(s, g, a^*) = 0.99^5$ and $Q^*(s, g, a_-) = 0.99^7$; while $C^*(s, a^*, g, h^*) = 1$ and $C^*(s, a_-, g, h^*) = 0$.

We propose to address these limitations by learning horizon-aware policies $\pi(a|s, g, h)$, which should be followed to reach goal $g$ from state $s$ in at most $h$ steps. The introduction of a time horizon $h$ naturally allows us to tune the speed/reliability trade-off, as an agent wishing to reach the goal faster should select a policy with a suitably small $h$ value. To learn these policies, we introduce the *optimal cumulative accessibility function* $C^*(s, a, g, h)$. This is a generalization of the state-action value function and corresponds to the probability of reaching goal $g$ from state $s$ after at most $h$ steps if action $a$ is taken, and the agent acts optimally thereafter. Intuitively it is similar to the optimal $Q$-function, but $Q$-functions rarely correspond to probabilities, whereas the $C^*$-function does so by construction. We derive Bellman backup update rules for $C^*$, which allow it to be learned via minimization of unbiased estimates of the cross-entropy loss – this is in contrast to $Q$-learning, which optimizes biased estimates of the squared error. Policies $\pi(a|s, g, h)$ can then be recovered from the $C^*$ function. We call our method cumulative accessibility estimation, or *C-learning*. Pong et al. (2018) proposed TDMs, a method involving horizon-aware policies. We point out that their method is roughly related to a non-cumulative version of ours with a different loss that does not enable the speed/reliability trade-off and is ill-suited for sparse rewards. We include a detailed discussion of TDMs in section 4.

One might expect that adding an extra dimension to the learning task, namely $h$, would increase the difficulty - as $C^*$ effectively contains the information of several optimal $Q$-functions for different discount factors. However, we argue that $C^*$ does not need to be learned to the same degree of accuracy as the optimal $Q$-function for the agent to solve the task. As a result, learning $C^*$ is more efficient, and converges in fewer environmental interactions. This property, combined with our proposed goal sampling technique and replay buffer used during training, provides empirical improvements over $Q$-function based methods.

In addition to these advantages, learning $C^*$ is itself useful, containing information that the horizon-aware policies do not. It estimates whether a goal $g$ is reachable from the current state $s$ within $h$ steps. In contrast, $\pi(a|s, g, h)$ simply returns some action, even for unreachable goals. We show that $C^*$ can be used to determine reachability with examples in a nonholonomic environment.

**Summary of contributions**: $(i)$ introducing $C$-functions and cumulative accessibility estimation for both discrete and continuous action spaces; $(ii)$ highlighting the importance of the speed vs reliability trade-off in finite horizon reinforcement learning; $(iii)$ introducing a novel replay buffer specially tailored for learning $C^*$ which builds on HER (Andrychowicz et al., 2017); and $(iv)$ empirically showing the effectiveness of our method for goal-reaching as compared to existing alternatives, particularly in the context of complex motion planning tasks.

## 2 BACKGROUND AND RELATED WORK

Let us extend the Markov Decision Process (MDP) formalism (Sutton et al., 1998) for goal-reaching. We consider a set of actions $\mathcal{A}$, a state space $\mathcal{S}$, and a goal set $\mathcal{G}$. We assume access to a goal checking function $G : \mathcal{S} \times \mathcal{G} \rightarrow \{0, 1\}$ such that $G(s, g) = 1$ if and only if state $s$ achieves goal $g$. For example, achieving the goal could mean exactly reaching a certain state, in which case $\mathcal{G} = \mathcal{S}$ and

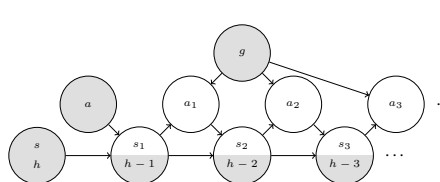

Figure 2: Graphical model depicting trajectories from $\mathbb{P}_{\pi(\cdot|\cdot,g,h)}(\cdot|s_0 = s, a_0 = a)$. Gray nodes denote fixed values, and white nodes stochastic ones. Nodes $a$, $g$ and $s$ are non-stochastic simply because they are conditioned on, not because they are always fixed within the environment. Note that the values of $h$ decrease deterministically. Nodes corresponding to horizons could be separated from states, but are not for a more concise graph.

$G(s, g) = \mathbb{1}(s = g)$. For many continuous state-spaces, hitting a state exactly has zero probability. Here we can still take $\mathcal{G} = \mathcal{S}$, but let $G(s, g) = \mathbb{1}(d(s, g) \leq \epsilon)$ for some radius $\epsilon$ and metric $d$. More general choices are possible. For example, in the Dubins' Car environment which we describe in more detail later, the state consists of both the location and orientation of the car: $\mathcal{S} = \mathbb{R}^2 \times S^1$. We take $\mathcal{G} = \mathbb{R}^2$, and $G(s, g)$ checks that the location of the car is within some small radius of $g$, ignoring the direction entirely. For a fixed $g$, $G(s, g)$ can be thought of as a sparse reward function.

In the goal-reaching setting, a policy $\pi : \mathcal{S} \times \mathcal{G} \to \mathcal{P}(\mathcal{A})$, where $\mathcal{P}(\mathcal{A})$ denotes the set of distributions over $\mathcal{A}$, maps state-goal pairs to an action distribution. The environment dynamics are given by a starting distribution $p(s_0, g)$, usually taken as $p(s_0)p(g)$, and transition probabilities $p(s_{t+1}|s_t, a_t)$. States for which $G(s, g) = 1$ are considered terminal.

**Q-Learning:** A $Q$-function (Watkins & Dayan, 1992) for multi-goal reaching, $Q^\pi : \mathcal{S} \times \mathcal{G} \times \mathcal{A} \to \mathbb{R}$, is defined by $Q^\pi(s_t, g, a_t) = \mathbb{E}_\pi[\sum_{i=t}^\infty \gamma^{i-t} G(s_t, g)|s_t, a_t]$, where $\gamma \in [0, 1]$ is a discount factor and the expectation is with respect to state-action trajectories obtained by using $\pi(a|s_i, g)$. If $\pi^*$ is an optimal policy in the sense that $Q^{\pi^*}(s, g, a) \geq Q^\pi(s, g, a)$ for every $\pi$ and $(s, g, a) \in \mathcal{S} \times \mathcal{G} \times \mathcal{A}$, then $Q^{\pi^*}$ matches the optimal $Q$-function, $Q^*$, which obeys the Bellman equation:

$$Q^*(s, g, a) = \mathbb{E}_{s' \sim p(\cdot|s,a)} \left[ G(s, g) + \gamma \max_{a' \in \mathcal{A}} Q^*(s', g, a') \right]. \tag{1}$$

In deep $Q$-learning (Mnih et al., 2015), $Q^*$ is parameterized with a neural network and learning is achieved by enforcing the relationship from equation 1. This is done by minimizing $\sum_i \mathcal{L}(Q^*(s_i, g_i, a_i), y_i)$, where $y_i$ corresponds to the expectation in equation 1 and is estimated using a replay buffer of stored tuples $(s_i, a_i, g_i, s_i')$. Note that $s_i'$ is the state the environment transitioned to after taking action $a_i$ from state $s_i$, and determines the value of $y_i$. Typically $\mathcal{L}$ is chosen as a squared error loss, and the dependency of $y_i$ on $Q^*$ is ignored for backpropagation in order to stabilize training. Once $Q^*$ is learned, the optimal policy is recovered by $\pi^*(a|s, g) = \mathbb{1}(a = \arg\max_{a'} Q^*(s, g, a'))$.

There is ample work extending and improving upon deep $Q$-learning (Haarnoja et al., 2018). For example, Lillicrap et al. (2015) extend it to the continuous action space setting, and Fujimoto et al. (2018) further stabilize training. These improvements are fully compatible with goal-reaching (Pong et al., 2019; Bharadhwaj et al., 2020a; Ghosh et al., 2019). Andrychowicz et al. (2017) proposed Hindsight Experience Replay (HER), which relabels past experience as achieved goals, and allows sample efficient learning from sparse rewards (Nachum et al., 2018).

## 3 CUMULATIVE ACCESSIBILITY FUNCTIONS

We now consider horizon-aware policies $\pi : \mathcal{S} \times \mathcal{G} \times \mathbb{N} \to \mathcal{P}(\mathcal{A})$, and define the cumulative accessibility function $C^\pi(s, a, g, h)$, or $C$-function, as the probability of reaching goal $g$ from state $s$ in at most $h$ steps by taking action $a$ and following the policy $\pi$ thereafter. By "following the policy $\pi$ thereafter" we mean that after $a$, the next action $a_1$ is sampled from $\pi(\cdot|s_1, g, h-1)$, $a_2$ is sampled from $\pi(\cdot|s_2, g, h-2)$ and so on. See Figure 2 for a graphical model depiction of how these trajectories are obtained. Importantly, an agent need not always act the same way at a particular state in order to reach a particular goal, thanks to horizon-awareness. We use $\mathbb{P}_{\pi(\cdot|\cdot,g,h)}(\cdot|s_0 = s, a_0 = a)$ to denote probabilities in which actions are drawn in this manner and transitions are drawn according to the environment $p(s_{t+1}|s_t, a)$. More formally, $C^\pi$ is given by:

$$C^\pi(s, a, g, h) = \mathbb{P}_{\pi(\cdot|\cdot,g,h)} \left( \max_{t=0,\ldots,h} G(s_t, g) = 1 \middle| s_0 = s, a_0 = a \right). \tag{2}$$

**Proposition 1**: $C^\pi$ can be framed as a $Q$-function within the MDP formalism, and if $\pi^*$ is optimal in the sense that $C^{\pi^*}(s, a, g, h) \geq C^\pi(s, a, g, h)$ for every $\pi$ and $(s, a, g, h) \in \mathcal{S} \times \mathcal{A} \times \mathcal{G} \times \mathbb{N}$, then $C^{\pi^*}$ matches the optimal $C$-function, $C^*$, which obeys the following equation:

$$C^*(s, a, g, h) = \begin{cases} \mathbb{E}_{s' \sim p(\cdot|s,a)} \left[ \max_{a' \in \mathcal{A}} C^*(s', a', g, h-1) \right] & \text{if } G(s,g) = 0 \text{ and } h \geq 1, \\ G(s,g) & \text{otherwise.} \end{cases} \tag{3}$$

See appendix A for a detailed mathematical proof of this proposition. The proof proceeds by first deriving a recurrence relationship that holds for any $C^\pi$. In an analogous manner to the Bellman equation in $Q$-learning, this recurrence involves an expectation over $\pi(\cdot|s', g, h-1)$, which, when replaced by a $\max$ returns the recursion for $C^*$.

Proposition 1 is relevant as it allows us to learn $C^*$, enabling goal-reaching policies to be recovered:

$$\pi^*(a|s, g, h) = \mathbb{1} \left( a = \arg\max_{a'} C^*(s, a', g, h) \right). \tag{4}$$

$C^*$ itself is useful for determining reachability. After maximizing over actions, it estimates whether a given goal is reachable from a state within some horizon. Comparing these probabilities for different horizons allows us to make a speed / reliability trade-off for reaching goals.

We observe that an *optimal* $C^*$-function is non-decreasing in $h$, but this does not necessarily hold for *non-optimal* $C$-functions. For example, a horizon-aware policy could actively try to avoid the goal for high values of $h$, and the $C^\pi$-function constructed from it would show lower probabilities of success for larger $h$. See appendix A for a concrete example of this counter-intuitive behavior.

**Proposition 2**: $C^*$ is non-decreasing in $h$.
See appendix A for a detailed mathematical proof. Intuitively, the proof consists of showing that an optimal policy can not exhibit the pathology mentioned above. Given an optimal policy $\pi^*(a|s, g, h)$ for a fixed horizon $h$ we construct a policy $\tilde{\pi}$ for $h+1$ which always performs better, and lower bounds the performance of $\pi^*(a|s, g, h+1)$.

In addition to being an elegant theoretical property, proposition 2 suggests that there is additional structure in a $C^*$ function which mitigates the added complexity from using horizon-aware policies. Indeed, in our preliminary experiments we used a non-cumulative version of $C$-functions (see section 3.3) and obtained significantly improved performance upon changing to $C$-functions. Moreover, monotonicity in $h$ could be encoded in the architecture of $C^*$ (Sill, 1998; Wehenkel & Louppe, 2019). However, we found that actively doing so hurt empirical performance (appendix F).

### 3.1 Shortcomings of $Q$-learning

Before describing our method for learning $C^*$, we highlight a shortcoming of $Q$-learning. Consider a 2D navigation environment where an agent can move deterministically in the cardinal directions, and fix $s$ and $g$. For an optimal action $a^*$, the optimal $Q$ function will achieve some value $Q^*(s, g, a^*) \in [0, 1]$ in the sparse reward setting. Taking a sub-optimal action $a^-$ initially results in the agent taking two extra steps to reach the intended goal, given that the agent acts optimally after the first action, so that $Q^*(s, g, a^-) = \gamma^2 Q^*(s, g, a^*)$. The value of $\gamma$ is typically chosen close to 1, for example 0.99, to ensure that future rewards are not too heavily discounted. As a consequence $\gamma^2 \approx 1$ and thus the value of $Q^*$ at the optimal action is very close to its value at a sub-optimal action. We illustrate this issue in Figure 1b. In this scenario, recovering an optimal policy requires that the error between the learned $Q$-function and $Q^*$ should be at most $(1 - \gamma^2)/2$; this is reflected empirically by $Q$-learning having high sample complexity and learning sub-optimal paths. This shortcoming surfaces in any environment where taking a sub-optimal action results in a slightly longer path than an optimal one, as in e.g. motion planning tasks.

The $C^*$ function does not have this shortcoming. Consider the same 2D navigation example, and let $h^*$ be the smallest horizon for which $g$ can be reached from $s$. $h^*$ can be easily obtained from $C^*$ as the smallest $h$ such that $\max_a C^*(s, a, g, h) = 1$. Again, denoting $a^*$ as an optimal action and $a^-$ as a sub-optimal one, we have that $C^*(s, a^*, g, h^*) = 1$ whereas $C^*(s, a^-, g, h^*) = 0$, which is illustrated in Figure 1b. Therefore, the threshold for error is much higher when learning the $C^*$ function. This property results in fewer interactions with the environment needed to learn $C^*$ and more efficient solutions.

### 3.2 Horizon-independent policies

Given a $C^*$-function, equation 4 lets us recover a horizon-aware policy. At test time, using small values of $h$ can achieve goals faster, while large values of $h$ will result in safe policies. However, a natural question arises: how small is small and how large is large? In this section we develop a method to quantitatively recover reasonable values of $h$ which adequately balance the speed/reliability trade-off. First, a safety threshold $\alpha \in (0, 1]$ is chosen, which indicates the percentage of the maximum value of $C^*$ we are willing to accept as safe enough. Smaller values of $\alpha$ will thus result in quicker policies, while larger values will result in safer ones. Then we consider a range of viable horizons, $\mathcal{H}$, and find the maximal $C^*$ value, $M(s, g) = \max_{h \in \mathcal{H}, a \in \mathcal{A}} C^*(s, a, g, h)$. We then compute:

$$h_\alpha(s, g) = \arg\min_{h \in \mathcal{H}} \{\max_{a \in \mathcal{A}} C^*(s, a, g, h) : \max_{a \in \mathcal{A}} C^*(s, a, g, h) \geq \alpha M(s, g)\}, \tag{5}$$

and take $\pi_\alpha^*(a|s, g) = \mathbb{1}\,(a = \arg\max_{a'} C^*(s, a', g, h_\alpha(s, g)))$. This procedure also allows us to recover horizon-independent policies from $C^*$ by using a fixed value of $\alpha$, which makes comparing against horizon-unaware methods straightforward. We used horizon-unaware policies with added randomness as the behaviour policy when interacting with the environment for exploration.

### 3.3 Alternative recursions

One could consider defining a non-cumulative version of the $C$-function, which we call an $A$-function for "accessibility" (*not* for "advantage"), yielding the probability of reaching a goal in exactly $h$ steps. However, this version is more susceptible to pathological behaviors that hinder learning for certain environments. For illustration, consider an agent that must move one step at a time in the cardinal directions on a checkerboard. Starting on a dark square, the probability of reaching a light square in an even number of steps is always zero, but may be non-zero for odd numbers. An optimal $A$-function would fluctuate wildly as the step horizon $h$ is increased, resulting in a harder target to learn. Nonetheless, $A$-functions admit a similar recursion to equation 18, which we include in appendix C for completeness. In any case, the $C$-function provides a notion of reachability which is more closely tied to related work (Savinov et al., 2018; Venkattaramanujam et al., 2019; Ghosh et al., 2018; Bharadhwaj et al., 2020a).

In $Q$-learning, discount factors close to 1 encourage safe policies, while discount factors close to 0 encourage fast policies. One could then also consider discount-conditioned policies $\pi(a|s, g, \gamma)$ as a way to achieve horizon-awareness. In appendix D we introduce $D$-functions (for "discount"), $D(s, a, g, \gamma)$, which allow recovery of discount-conditioned policies. However $D$-functions suffer from the same limitation as $Q$-functions in that they need to be learned to a high degree of accuracy.

## 4 Cumulative accessibility estimation

Our training algorithm, which we call cumulative accessibility estimation (CAE), or $C$-learning, is detailed in algorithm 1. Similarly to $Q$-learning, the $C^*$ function can be modelled as a neural network with parameters $\theta$, denoted $C_\theta^*$, which can be learned by minimizing:

$$-\sum_i \left[y_i \log C_\theta^*(s_i, a_i, g_i, h_i) + (1 - y_i) \log\left(1 - C_\theta^*(s_i, a_i, g_i, h_i)\right)\right], \tag{6}$$

where $y_i$ corresponds to a stochastic estimate of the right hand side of equation 18. The sum is over tuples $(s_i, a_i, s_i', g_i, h_i)$ drawn from a replay buffer which we specially tailor to successfully learn $C^*$. Since $C^*$ corresponds to a probability, we change the usual squared loss to be the binary cross entropy loss. Note that even if the targets $y_i$ are not necessarily binary, the use of binary cross entropy is still justified as it is equivalent to minimizing the $\mathbb{KL}$ divergence between Bernoulli distributions with parameters $y_i$ and $C_\theta^*(s_i, a_i, g_i, h_i)$ (Bellemare et al., 2017). Since the targets used do not correspond to the right-hand side of equation 18 but to a stochastic estimate (through $s_i'$) of it, using this loss instead of the square loss results in an *unbiased* estimate of $\sum_i \mathcal{L}(C_\theta^*(s_i, a_i, g_i, h_i), y_i^{\text{true}})$, where $y_i^{\text{true}}$ corresponds to the right-hand side of equation 18 at $(s_i, a_i, g_i, h_i)$ (see appendix B for a detailed explanation). This confers another benefit of $C$-learning over $Q$-learning, as passing a stochastic estimate of equation 1 through the typical squared loss results in *biased* estimators. As in Double $Q$-learning (Van Hasselt et al., 2015), we use a second network $C_{\theta'}^*$ to compute the $y_i$ targets, and do not minimize equation 6 with respect to $\theta'$. We periodically update $\theta'$ to match $\theta$. We now explain our algorithmic details.

---

**Algorithm 1:** Training C-learning Network

---

**Parameter:** $N_{\text{explore}}$: Number of random exploration episodes
**Parameter:** $N_{\text{GD}}$: Number of goal-directed episodes
**Parameter:** $N_{\text{train}}$: Number of batches to train on per goal-directed episode
**Parameter:** $N_{\text{copy}}$: Number of batches between target network updates
**Parameter:** $\alpha$: Learning rate

1  $\theta \leftarrow$ Initial weights for $C_\theta^*$
2  $\theta' \leftarrow \theta$                                  // Copy weights to target network
3  $\mathcal{D} \leftarrow [\,]$                                // Initialize experience replay buffer
4  $n_b \leftarrow 0$                                          // Counter for training batches
5  **repeat** $N_{\text{explore}}$ **times**
6  $\quad \mathcal{E} \leftarrow \texttt{get\_rollout}(\pi_{\text{random}})$                    // Do a random rollout
7  $\quad \mathcal{D}.\texttt{append}(\mathcal{E})$                         // Save the rollout to the buffer
8  **repeat** $N_{\text{GD}}$ **times**
9  $\quad g \leftarrow \texttt{goal\_sample}(n_b)$                          // Sample a goal
10 $\quad \mathcal{E} \leftarrow \texttt{get\_rollout}(\pi_{\text{behavior}})$                 // Try to reach the goal
11 $\quad \mathcal{D}.\texttt{append}(\mathcal{E})$                         // Save the rollout
12 $\quad$ **repeat** $N_{\text{train}}$ **times**
13 $\quad\quad$ **if** $n_b \mod N_{\text{copy}} = 0$ **then**
14 $\quad\quad\quad \theta' \leftarrow \theta$                             // Copy weights periodically
15 $\quad\quad \mathcal{B} := \{s_i, a_i, s_i', g_i, h_i\}_{i=1}^{|\mathcal{B}|} \leftarrow \texttt{sample\_batch}(\mathcal{D})$        // Sample a batch
16 $\quad\quad \{y_i\}_{i=1}^{|\mathcal{B}|} \leftarrow \texttt{get\_targets}(\mathcal{B}, \theta')$          // Estimate RHS of equation 18
17 $\quad\quad \hat{\mathcal{L}} \leftarrow -\frac{1}{|\mathcal{B}|} \sum_{i=1}^{|\mathcal{B}|} y_i \log C_\theta^*(s_i, a_i, g_i, h_i) + (1 - y_i) \log (1 - C_\theta^*(s_i, a_i, g_i, h_i))$
18 $\quad\quad \theta \leftarrow \theta - \alpha \nabla_\theta \hat{\mathcal{L}}$                          // Update weights
19 $\quad\quad n_b \leftarrow n_b + 1$                                     // Trained one batch

---

**Reachability-guided sampling**: When sampling a batch (line 15 of algorithm 1), for each $i$ we first sample an episode from $\mathcal{D}$, and then a transition $(s_i, a_i, s_i')$ from that episode. We sample $h_i$ favoring smaller $h$'s at the beginning of training. We achieve this by selecting $h_i = h$ with probability proportional to $h^{-\kappa n_{GD}/N_{GD}}$, where $n_{GD}$ is the current episode number, $N_{GD}$ the total number of episodes, and $\kappa$ is a hyperparameter. For deterministic environments, we follow HER (Andrychowicz et al., 2017) and select $g_i$ uniformly at random from the states observed in the episode after $s_i$. For stochastic environments we use a slightly modified version, which addresses the bias incurred by HER (Matthias et al., 2018) in the presence of stochasticity. All details are included in appendix G.

**Extension to continuous action spaces**: Note that constructing the targets $y_i$ requires taking a maximum over the action space. While this is straightforward in discrete action spaces, it is not so for continuous actions. Lillicrap et al. (2015) proposed DDPG, a method enabling deep $Q$-learning in continuous action spaces. Similarly, Fujimoto et al. (2018) proposed TD3, a method for further stabilizing $Q$-learning. We note that $C$-learning is compatible with the ideas underpinning both of these methods. We present a TD3 version of $C$-learning in appendix E.

We finish this section by highlighting differences between $C$-learning and related work. Ghosh et al. (2019) proposed GCSL, a method for goal-reaching inspired by supervised learning. In their derivations, they also include a horizon $h$ which their policies can depend on, but they drop this dependence in their experiments as they did not see a practical benefit by including $h$. We find the opposite for $C$-learning. TDMs (Pong et al., 2018) use horizon-aware policies and a similar recursion to ours. In practice they use a negative $L_1$ distance reward, which significantly differs from our goal-reaching indicators. This is an important difference as TDMs operate under dense rewards, while we use sparse rewards, making the problem significantly more challenging. Additionally, distance between states in nonholonomic environments is very poorly described by $L_1$ distance, resulting in TDMs being ill-suited for motion planning tasks. We also highlight that even if the reward in TDMs was swapped with our goal checking function $G$, the resulting objective would be much closer to the non-cumulative version of $C$-learning presented in appendix C. TDMs recover policies from their horizon-aware $Q$-function in a different manner to ours, not allowing a speed vs reliability trade-off. The recursion presented for TDMs is used by definition, whereas we have

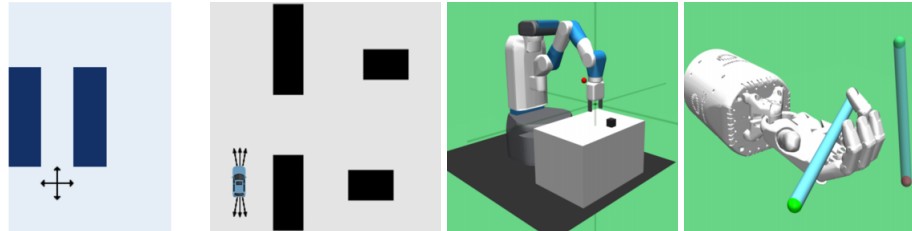

Figure 3: Experimental environments, from left to right: frozen lake, Dubins' car, FetchPickAndPlace-v1 and HandManipulatePenFull-v0. Arrows represent actions (only their direction should be considered, their magnitude is not representative of the distance travelled by an agent taking an action). See text for description.

provided a mathematical derivation of the Bellman equation for $C$-functions along with proofs of additional structure. As a result of the $C$-function being a probability we use a different loss function (binary cross-entropy) which results in unbiased estimates of the objective's gradients. Finally, we point out that TDMs sample horizons uniformly at random, which differs from our specifically tailored replay buffer.

## 5 EXPERIMENTS

### 5.1 SETUP

Our experiments are designed to show that $(i)$ $C$-learning enables the speed vs reliability trade-off, $(ii)$ the $C^*$ function recovered through $C$-learning meaningfully matches reachability in nonholonomic environments, and $(iii)$ $C$-learning scales to complex and high-dimensional motion planning tasks, resulting in improved sample complexity and goal-reaching. We use success rate (percentage of trajectories which reach the goal) and path length (average, over trajectories and goals, number of steps needed to achieve the goal among successful trajectories) as metrics.

We compare $C$-learning against GCSL (Ghosh et al., 2019) (for discrete action states only) and deep $Q$-learning with HER (Andrychowicz et al., 2017) across several environments, which are depicted in Figure 3. All experimental details and hyperparameters are given in appendix G. We also provide ablations, and comparisons against TDMs and the alternative recursions from section 3.3 in appendix F. We evaluate $C$-learning on the following domains:

1. **Frozen lake** is a 2D navigation environment where the agent must navigate without falling in the holes (dark blue). Both the state space ($7 \times 5$ grid, with two $3 \times 1$ holes) and the action state are discrete. Falling in a hole terminates an episode. The agent's actions correspond to intended directions. The agent moves in the intended direction with probability $0.8$, and in perpendicular directions with probabilities $0.1$ each. Moving against the boundaries of the environment has no effect. We take $\mathcal{G} = \mathcal{S}$ and $G(s, g) = \mathbb{1}(g = s)$.

2. **Dubins' car** is a more challenging deterministic 2D navigation environment, where the agent drives a car which cannot turn by more than $10°$. The states, with spatial coordinates in $[0, 15]^2$, are continuous and include the direction of the car. There are 7 actions: the 6 combinations of $\{\text{left } 10°, \text{ straight}, \text{ right } 10°\} \times \{\text{forward } 1, \text{ reverse } 1\}$, and the option to not move. As such, the set of reachable states is not simply a ball around the current state. The environment also has walls through which the car cannot drive. We take $\mathcal{G} = [0, 15]^2$ and the goal is considered to be reached when the car is within an $L_\infty$ distance of $0.5$ from the goal, regardless of its orientation.

3. **FetchPickAndPlace-v1** (Brockman et al., 2016) is a complex, higher-dimensional environment in which a robotic arm needs to pick up a block and move it to the goal location. Goals are defined by their 3-dimensional coordinates. The state space is 25-dimensional, and the action space is continuous and 4-dimensional.

4. **HandManipulatePenFull-v0** (Brockman et al., 2016) is a realistic environment known the be a difficult goal-reaching problem, where deep Q-learning with HER shows very limited success (Plappert et al., 2018). The environment has a continuous action space of dimension 20, a 63-dimensional state space, and 7-dimensional goals. Note that we are considering the more challenging version of the environment where both target location and orientation are chosen randomly.

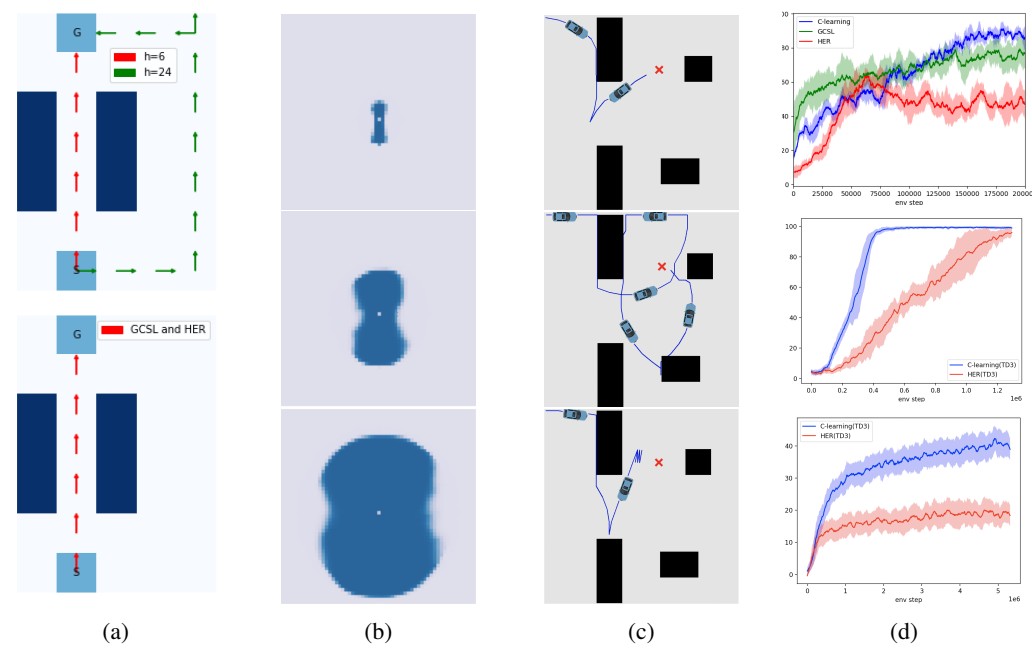

$$(a) \qquad\qquad (b) \qquad\qquad (c) \qquad\qquad (d)$$

Figure 4: $(a)$ Multimodal policies recovered by $C$-learning in frozen lake for different values of $h$ for reaching $(G)$ from $(S)$ (top); unimodal policies recovered by GCSL and HER (bottom). $(b)$ Heatmaps over the goal space of $\max_a C^*(s, a, g, h)$ with a fixed $s$ and $h$ for Dubins' car, with $h = 7$ (top), $h = 15$ (middle) and $h = 25$ (bottom). $(c)$ Trajectories learned by $C$-learning (top), GCSL (middle) and HER (bottom) for Dubins' car. $(d)$ Success rate throughout the training for Dubins' car (top), FetchPickAndPlace-v1 (middle) and HandManipulatePenFull-v0 (bottom) for $C$-learning (blue), HER (red) and GCSL (green).

## 5.2 RESULTS

**Speed vs reliability trade-off**: We use frozen lake to illustrate this trade-off. At test time, we set the starting state and goal as shown in Figure 4a. Notice that, given enough time, an agent can reach the goal with near certainty by going around the holes on the right side of the lake. However, if the agent is time-constrained, the optimal policy must accept the risk of falling in a hole. We see that $C$-learning does indeed learn both the risky and safe policies. Other methods, as previously explained, can only learn one. To avoid re-plotting the environment for every horizon, we have plotted arrows in Figure 4a corresponding to $\arg\max_a \pi^*(a|s_0, g, h)$, $\arg\max_a \pi^*(a|s_1, g, h-1)$, $\arg\max_a \pi^*(a|s_2, g, h-2)$ and so on, where $s_{t+1}$ is obtained from the previous state and action while ignoring the randomness in the environment (i.e. $s_{t+1} = \arg\max_s p(s|s_t, a_t)$). In other words, we are plotting the most likely trajectory of the agent. When given the minimal amount of time to reach the goal ($h = 6$), the CAE agent learns correctly to accept the risk of falling in a hole by taking the direct path. When given four times as long ($h = 24$), the agent takes a safer path by going around the hole. Notice that the agent makes a single mistake at the upper right corner, which is quickly corrected when the value of $h$ is decreased. On the other hand, we can see on the bottom panel that both GCSL and HER recover a single policy, thus not enabling the speed vs reliability trade-off. Surprisingly, GCSL learns to take the high-risk path, despite Ghosh et al. (2019)'s intention to incentivize paths that are guaranteed to reach the goal.

**Reachability learning**: To demonstrate that we are adequately learning reachability, we removed the walls from Dubins' car and further restricted the turning angles to $5°$, thus making the dumbbell shape of the true reachable region more extreme. In Figure 4b, we show that our learned $C^*$ function correctly learns which goals are reachable from which states for different time horizons in Dubins' car: not only is the learned $C^*$ function increasing in $h$, but the shapes are as expected. None of the competing alternatives recover this information in any way, and thus comparisons are not available. As previously mentioned, the optimal $C^*$-function in this task is not trivial. Reachability is defined by the "geodesics" in $\mathcal{S} = [0, 15]^2 \times S^1$ that are constrained by the finite turning radius, and thus follow a more intricate structure than a ball in $\mathbb{R}^2$.

Table 1: Comparison of $C$-Learning against relevant benchmarks in three environments, averaged across five random seeds. Runs in **bold** are either the best on the given metric in that environment, or have a mean score within the error bars of the best.

| ENVIRONMENT | METHOD | SUCCESS RATE | PATH LENGTH |
|---|---|---|---|
| Dubins' Car | CAE | $\mathbf{86.15\% \pm 2.44\%}$ | $\mathbf{16.45 \pm 0.99}$ |
| | GCSL | $79.69\% \pm 6.35\%$ | $32.64 \pm 6.16$ |
| | HER | $51.25\% \pm 6.48\%$ | $20.13 \pm 1.66$ |
| FetchPickAndPlace-v1 | CAE (TD3) | $\mathbf{99.34\% \pm 0.27\%}$ | $\mathbf{8.53 \pm 0.09}$ |
| | HER (TD3) | $98.25\% \pm 2.51\%$ | $8.86 \pm 0.14$ |
| HandManipulatePenFull-v0 | CAE (TD3) | $\mathbf{39.83\% \pm 1.64\%}$ | $\mathbf{7.68 \pm 0.90}$ |
| | HER (TD3) | $21.99\% \pm 2.46\%$ | $15.03 \pm 0.71$ |

**Motion planning and goal-reaching**: For a qualitative comparison of performance for goal-reaching, Figure 4c shows trajectories for $C$-learning and competing alternatives for Dubins' car. We can see that $C$-learning finds the optimal route, which is a combination of forward and backward movement, while other methods find inefficient paths. We also evaluated our method on challenging goal reaching environments, and observed that $C$-learning achieves state-of-the-art results both in sample complexity and success rate. Figure 4d shows that $C$-learning is able to learn considerably faster in the FetchPickAndPlace-v1 environment. More importantly, $C$-learning achieves an absolute $20\%$ improvement in its success rate over the current state-of-the-art algorithm HER (TD3) in the HandManipulatePenFull-v0 environment. Quantitative results are shown in Table 1. On Dubins' car, we also note that $C$-learning ends up with a smaller final $L_\infty$ distance to the goal: $\mathbf{0.93 \pm 0.15}$ vs. $1.28 \pm 0.51$ for GCSL.

## 6 DISCUSSION

We have shown that $C$-learning enables simultaneous learning of how to reach goals quickly and how to reach them safely, which current methods cannot do. We point out that reaching the goal safely in our setting means doing so at test time, and is different to what is usually considered in the safety literature where safe exploration is desired (Chow et al., 2018; Bharadhwaj et al., 2020b). Additionally, learning $C^*$ effectively learns reachability within an environment, and could thus naturally lends itself to incorporation into other frameworks, for example, in goal setting for hierarchical RL tasks (Nachum et al., 2018) where intermediate, reachable goals need to be selected sequentially. We believe further investigations on using $C$-functions on safety-critical environments requiring adaptation (Peng et al., 2018; Zhang et al., 2020), or for hierarchical RL, are promising directions for further research.

We have also argued that $C$-functions are more tolerant of errors during learning than $Q$-functions which increases sample efficiency. This is verified empirically in that $C$-learning is able to solve goal-reaching tasks earlier on in training than $Q$-learning.

We finish by noticing a parallel between $C$-learning and the options framework (Sutton et al., 1999; Bacon et al., 2017), which introduces temporal abstraction and allows agents to not always follow the same policy when at a given state $s$. However, our work does not fit within this framework, as options evolve stochastically and new options are selected according only to the current state, while horizons evolve deterministically and depend on the previous horizon only, not the state. Additionally, and unlike $C$-learning, nothing encourages different options to learn safe or quick policies, and there is no reachability information contained in options.

## 7 CONCLUSIONS

In this paper we introduced $C$-learning, a $Q$-learning-inspired method for goal-reaching. Unlike previous approaches, we propose the use of horizon-aware policies, and show that not only can these policies be tuned to reach the goal faster or more reliably, but they also outperform horizon-unaware approaches for goal-reaching on complex motion planning tasks. We hope our method will inspire further research into horizon-aware policies and their benefits.

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

# A    Proofs of propositions

**Proposition 1**: $C^\pi$ can be framed as a $Q$-function within the MDP formalism, and if $\pi^*$ is optimal in the sense that $C^{\pi^*}(s, a, g, h) \geq C^\pi(s, a, g, h)$ for every $\pi$ and $(s, a, g, h) \in \mathcal{S} \times \mathcal{A} \times \mathcal{G} \times \mathbb{N}$, then $C^{\pi^*}$ matches the optimal $C$-function, $C^*$, which obeys the following equation:

$$C^*(s, a, g, h) = \begin{cases} \mathbb{E}_{s' \sim p(\cdot|s,a)} \left[ \max_{a' \in \mathcal{A}} C^*(s', a', g, h-1) \right] & \text{if } G(s, g) = 0 \text{ and } h \geq 1, \\ G(s, g) & \text{otherwise.} \end{cases}$$

**Proof**: First, we frame $C$-functions within the MDP formalism. Consider a state space given by $\mathcal{S}' = \mathcal{S} \times \mathcal{G} \times \mathbb{N}$, where states corresponding to reached goals (i.e. $G(s, g) = 1$) or with last coordinate equal to 0 (i.e. $h = 0$) are considered terminal. The dynamics in this environment are given by:

$$p(s_{t+1}, g_{t+1}, h_{t+1} | s_t, g_t, h_t, a_t) = p(s_{t+1} | s_t, a_t) \mathbb{1}(g_{t+1} = g_t) \mathbb{1}(h_{t+1} = h_t - 1). \tag{7}$$

That is, states evolve according to the original dynamics, while goals remain unchanged and horizons decrease by one at every step. The initial distribution is given by:

$$p(s_0, g_0, h_0) = p(s_0) p(g_0, h_0 | s_0), \tag{8}$$

where $p(s_0)$ corresponds to the original starting distribution and $p(g_0, h_0|s_0)$ needs to be specified beforehand. Together, the initial distribution and the dynamics, along with a policy $\pi(a|s, g, h)$ properly define a distribution over trajectories, and again, we use $\mathbb{P}_{\pi(\cdot|\cdot,g,h)}$ to denote probabilities with respect to this distribution. The reward function $r : \mathcal{S}' \times \mathcal{A} \to \mathbb{R}$ is given by:

$$r(s, a, g, h) = G(s, g). \tag{9}$$

By taking the discount factor to be 1, and since we take states for which $G(s, g) = 1$ to be terminal, the return (sum of rewards) over a trajectory $(s_0, s_1, \dots)$ with $h_0 = h$ is given by:

$$\max_{t=0,\dots,h} G(s_t, g). \tag{10}$$

For notational simplicity we take $G(s_t, g) = 0$ whenever $t$ is greater than the length of the trajectory to properly account for terminal states. Since the return is binary, its expectation matches its probability of being equal to 1, so that indeed, the $Q$-functions in this MDP correspond to $C^\pi$:

$$C^\pi(s, a, g, h) = \mathbb{P}_{\pi(\cdot|\cdot,g,h)} \left( \max_{t=0,\dots,h} G(S_t, g) = 1 \middle| s_0 = s, a_0 = a \right).$$

Now, we derive the Bellman equation for our $C$-functions. Trivially:

$$C^\pi(s, a, g, h) = G(s, g), \tag{11}$$

whenever $G(s, g) = 1$ or $h = 0$. For the rest of the derivation, we assume that $G(s, g) = 0$ and $h \geq 1$. We note that the probability of reaching $g$ from $s$ in at most $h$ steps is given by the probability of reaching it in exactly one step, plus the probability of not reaching it in the first step and reaching it in at most $h - 1$ steps thereafter. Formally:

$$C^\pi(s, a, g, h) \tag{12}$$
$$= C^\pi(s, a, g, 1) + \mathbb{E}_{s' \sim p(\cdot|s,a)} \left[ (1 - G(s', g)) \mathbb{E}_{a' \sim \pi(\cdot|s',g,h-1)} [C^\pi(s', a', g, h-1)] \right]$$
$$= \mathbb{E}_{s' \sim p(\cdot|s,a)} [G(s', g)] + \mathbb{E}_{s' \sim p(\cdot|s,a)} \left[ (1 - G(s', g)) \mathbb{E}_{a' \sim \pi(\cdot|s',g,h-1)} [C^\pi(s', a', g, h-1)] \right]$$
$$= \mathbb{E}_{s' \sim p(\cdot|s,a)} \left[ \mathbb{E}_{a' \sim \pi(\cdot|s',g,h-1)} [C^\pi(s', a', g, h-1)] \right],$$

where the last equality follows from the fact that $C^\pi(s', a', g, h-1) = 1$ whenever $G(s', g) = 1$. Putting everything together, we obtain the Bellman equation for $C^\pi$:

$$C^\pi(s, a, g, h) = \begin{cases} \mathbb{E}_{s' \sim p(\cdot|s,a)} \left[ \mathbb{E}_{a' \sim \pi(\cdot|s',g,h-1)} [C^\pi(s', a', g, h-1)] \right] & \text{if } G(s, g) = 0 \text{ and } h \geq 1, \\ G(s, g) & \text{otherwise.} \end{cases} \tag{13}$$

Recall that the optimal policy is defined by the fact that $C^* \geq C^\pi$ for any arguments and for any policy $\pi$. We can by maximizing equation 13 that, given the $C^*(\cdot, \cdot, \cdot, h-1)$ values, the optimal policy values at horizon $h - 1$ must be $\pi^*(a|s, g, h-1) = \mathbb{1}(a = \arg\max_{a'} C^*(s, a, g, h-1))$.

We thus we obtain:

$$C^*(s, a, g, h) = \begin{cases} \mathbb{E}_{s' \sim p(\cdot|s,a)} \left[ \max_{a' \in \mathcal{A}} C^*(s', a', g, h-1) \right] & \text{if } G(s,g) = 0 \text{ and } h \geq 1, \\ G(s,g) & \text{otherwise.} \end{cases}$$

Note that, as in $Q$-learning, the Bellman equation for the optimal policy has been obtained by replacing the expectation with respect to $a'$ with a $\max$. □

**Proposition 2**: $C^*$ is non-decreasing in $h$.

As mentioned in the main manuscript, one might naively think that this monotonicity property should hold for $C^\pi$ for any $\pi$. However this is not quite the case, as $\pi$ depends on $h$ and a perverse policy may actively avoid the goal for large values of $h$. Restricting to optimal $C^*$-functions, such pathologies are avoided. As an example, consider an environment with three states, $\{0, 1, 2\}$, and two actions $\{-1, +1\}$. The transition rule is $s_{t+1} = \max(0, \min(2, s_t + a_t))$, that is, the agent moves deterministically in the direction of the action unless doing so would move it out of the domain. Goals are defined as states. Let $\pi$ be such that $\pi(a|s = 1, g = 2, h = 1) = \mathbb{1}(a = 1)$ and $\pi(a|s = 1, g = 2, h = 2) = \mathbb{1}(a = -1)$. While clearly a terrible policy, $\pi$ is such that $C^\pi(s = 0, a = 1, g = 2, h = 2) = 1$ and $C^\pi(s = 0, a = 1, g = 2, h = 3) = 0$, so that $C^\pi$ can decrease with $h$.

**Proof**: Fix a distribution $p$ on the action space. For any policy $\pi(a|s, g, h)$, we define a new policy $\tilde{\pi}$ as:

$$\tilde{\pi}(a|s, g, h+1) = \begin{cases} \pi(a|s, g, h), & \text{if } h > 0 \\ p(a) & , \text{ otherwise.} \end{cases} \tag{14}$$

The new policy $\tilde{\pi}$ acts the same as $\pi$ for the first $h$ steps and on the last step it samples an action from the fixed distribution $p$. The final step can only increase the cumulative probability of reaching the goal, therefore:

$$C^{\tilde{\pi}}(s, a, g, h+1) \geq C^\pi(s, a, g, h). \tag{15}$$

Since equation 15 holds for all policies $\pi$, taking the maximum over policies gives:

$$\begin{aligned} C^*(s, a, g, h+1) &\geq \max_\pi C^{\tilde{\pi}}(s, a, g, h+1) \\ &\geq \max_\pi C^\pi(s, a, g, h) \\ &= C^*(s, a, g, h). \end{aligned} \tag{16}$$

□

# B UNBIASEDNESS OF THE CROSS ENTROPY LOSS

For given $s_i$, $a_i$, $g_i$ and $h_i$ from the replay buffer, we denote the right-hand side of equation 2 as $y_i^{true}$:

$$y_i^{true} = \begin{cases} \mathbb{E}_{s' \sim p(\cdot|s_i,a_i)} \left[ \max_{a' \in \mathcal{A}} C_\theta^*(s', a', g_i, h_i - 1) \right] & \text{if } G(s_i, g_i) = 0 \text{ and } h_i \geq 1, \\ G(s_i, g_i) & \text{otherwise.} \end{cases} \tag{17}$$

Note that $y_i^{true}$ cannot be evaluated exactly, as the expectation over $s'$ would require knowledge of the environment dynamics to compute in closed form. However, using $s_i'$, the next state after $s_i$ in the replay buffer, we can obtain a single-sample Monte Carlo estimate of $y_i^{true}$, $y_i$ as:

$$y_i = \begin{cases} \max_{a' \in \mathcal{A}} C_\theta^*(s_i', a', g_i, h_i - 1) & \text{if } G(s_i, g_i) = 0 \text{ and } h_i \geq 1, \\ G(s_i, g_i) & \text{otherwise.} \end{cases} \tag{18}$$

Clearly $y_i$ is an unbiased estimate of $y_i^{true}$. However, the optimization objective is given by:

$$\sum_i \mathcal{L}(C_\theta^*(s_i, a_i, g_i, h_i), y_i^{true}) \tag{19}$$

where the sum is over tuples in the replay buffer and $\mathcal{L}$ is the loss being used. Simply replacing $y_i^{true}$ with its estimate $y_i$, while commonly done in $Q$-learning, will in general result in a biased estimate of the loss:

$$\sum_i \mathcal{L}(C_\theta^*(s_i, a_i, g_i, h_i), y_i^{true}) = \sum_i \mathcal{L}(C_\theta^*(s_i, a_i, g_i, h_i), \mathbb{E}_{s_i'|s_i, a_i}[y_i])$$

$$\neq \sum_i \mathbb{E}_{s_i'|s_i, a_i} \left[ \mathcal{L}(C_\theta^*(s_i, a_i, g_i, h_i), y_i) \right] \tag{20}$$

since in general the expectation of a function need not match the function of the expectation. In other words, pulling the expectation with respect to $s_i'$ outside of the loss will in general incur in bias. However, if $\mathcal{L}$ is linear in its second argument, as is the case with binary cross entropy but not with the squared loss, then one indeed recovers:

$$\sum_i \mathcal{L}(C_\theta^*(s_i, a_i, g_i, h_i), y_i^{true}) = \sum_i \mathbb{E}_{s_i'|s_i, a_i} \left[ \mathcal{L}(C_\theta^*(s_i, a_i, g_i, h_i), y_i) \right] \tag{21}$$

so that replacing $y_i^{true}$ with $y_i$ does indeed recover an unbiased estimate of the loss.

## C    Non-cumulative case

We originally considered a non-cumulative version of the $C$-function, giving the probability of reaching the goal in exactly $h$ steps. We call this function the accessibility function, $A^\pi$ (despite our notation, this is unrelated to the commonly used advantage function), defined by:

$$A^\pi(s, a, g, h) = \mathbb{P}_{\pi(\cdot|\cdot, g, h)} \left( G(s_h, g) = 1 | s_0 = s, a_0 = a \right). \tag{22}$$

Here the trivial case is only when $h = 0$, where we have:

$$A^\pi(s, a, g, 0) = G(s, g) \tag{23}$$

If $h \geq 1$, we can obtain a similar recursion to that of the $C$-functions. Here we no longer assume that states which reach the goal are terminal. The probability of reaching the goal in exactly $h$ steps is equal to the probability of reaching it in $h - 1$ steps *after* taking the first step. After the first action $a$, subsequent actions are sampled from the policy $\pi$:

$$
\begin{aligned}
A^\pi(s, a, g, h) &= \mathbb{P}_{\pi(\cdot|\cdot, g, h)} \left( G(s_h, g) = 1 | s_0 = s, a_0 = a \right) \\
&= \mathbb{E}_{s' \sim p(\cdot|s, a)} \left[ \mathbb{E}_{a' \sim \pi(\cdot|s', g, h-1)} \left[ \mathbb{P}_{\pi(\cdot|\cdot, g, h)} \left( G(s_h, g) = 1 | s_1 = s', a_1 = a' \right) \right] \right] \\
&= \mathbb{E}_{s' \sim p(\cdot|s, a)} \left[ \mathbb{E}_{a' \sim \pi(\cdot|s', g, h-1)} \left[ \mathbb{P}_{\pi(\cdot|\cdot, g, h-1)} \left( G(s_{h-1}, g) = 1 | s_0 = s', a_0 = a' \right) \right] \right] \\
&= \mathbb{E}_{s' \sim p(\cdot|s, a)} \left[ \mathbb{E}_{a' \sim \pi(\cdot|s', g, h-1)} \left[ A^\pi(s', a', g, h-1) \right] \right].
\end{aligned} \tag{24}
$$

By the same argument we employed in proposition 1, the recursion for the optimal $A$-function, $A^*$, is obtained by replacing the expectation with respect to $a'$ with max. Putting this together with the base case, we have:

$$A^*(s, a, g, h) = \begin{cases} \mathbb{E}_{s' \sim p(\cdot|s, a)} \left[ \max_{a' \in \mathcal{A}} A^*(s', a, g, h-1) \right] & \text{if } h \geq 1, \\ G(s, g) & \text{otherwise.} \end{cases} \tag{25}$$

Note that this recursion is extremely similar to that of $C^*$, and differs only in the base cases for the recursion. The difference is empirically relevant however for the two reasons that make $C^*$ easier to learn: $C^*$ is monotonic in $h$, and the cumulative probability is more well-behaved generally.

## D    Discounted case

Another approach which would allow a test-time trade-off between speed and reliability is to learn the following $D$-function ($D$ standing for "discounted accessibility").

$$D^\pi(s, a, g, \gamma) = \mathbb{E}_\pi \left[ \gamma^{T-1} | s_0 = s, a_0 = a \right], \tag{26}$$

where the random variable $T$ is the smallest positive number such that $G(s_T, g) = 1$. If no such state occurs during an episode then $\gamma^{T-1}$ is interpreted as zero. This is the discounted future return of an environment in which satisfying the goal returns a reward of 1 and terminates the episode.

We may derive a recursion relation for this formalism too

$$D^\pi(s, a, g, \gamma) = \mathbb{E}_{s' \sim p(\cdot|s,a)} \left[ G(s', g) + \gamma(1 - G(s', g))\mathbb{E}_{a' \sim \pi(\cdot|s')} \left[ \mathbb{E}_\pi \left[ \gamma^{T-2}|s', a' \right] \right] \right]$$

$$= \mathbb{E}_{s' \sim p(\cdot|s,a)} \left[ G(s', g) + \gamma(1 - G(s', g))\mathbb{E}_{a' \sim \pi(\cdot|s')} \left[ D^\pi(s', a', g, \gamma) \right] \right]. \quad (27)$$

By the same argument as employed for the $C$ and $A$ functions, the $D$-function of the optimal policy is obtained by replacing the expectation over actions with a max, giving

$$D^*(s, a, g, \gamma) = \mathbb{E}_{s' \sim p(\cdot|s,a)} \left[ G(s', g) + \gamma(1 - G(s', g)) \max_{a'} D^*(s', a', g, \gamma) \right]. \quad (28)$$

Learning such a $D$-function would allow a test time trade-off between speed and reliability, and might well be more efficient than training independent models for different values of $\gamma$. We did not pursue this experimentally for two reasons. Firstly, our initial motivation was to allow some higher-level operator (either human or a controlling program) to trade off speed for reliability, and a hard horizon is usually more interpretable than a discounting factor. Secondly, we noticed that $C$-learning performs strongly at goal-reaching in deterministic environments, and we attribute this to the hard horizon allowing the optimal policy to be executed even with significant errors in the $C$-value. Conversely, $Q$-learning can typically only tolerate small errors before the actions selected become sub-optimal. $D$-learning would suffer from the same issue.

## E  CONTINUOUS ACTION SPACE CASE

As mentioned in the main manuscript, $C$-learning is compatible with the ideas that underpin DDPG and TD3 (Lillicrap et al., 2015; Fujimoto et al., 2018), allowing it to be used in continuous action spaces. This requires the introduction of another neural network approximating a deterministic policy, $\mu_\phi : \mathcal{S} \times \mathcal{G} \times \mathbb{N} \to \mathcal{A}$, which is intended to return $\mu(s, g, h) = \arg \max_a C^*(s, a, g, h)$. This network is trained alongside $C_\theta^*$ as detailed in Algorithm 2.

We do point out that the procedure to obtain a horizon-agnostic policy, and thus $\pi_{\text{behavior}}$, is also modified from the discrete version. Here, $M(s, g) = \max_{h \in \mathcal{H}} C^*(s, \mu(s, g, h), g, h)$, and:

$$h_\gamma(s, g) = \arg \min_{h \in \mathcal{H}} \{ C^*(s, \mu(s, g, h), g, h) : C^*(s, \mu(s, g, h), g, h) \geq \gamma M(s, g) \}, \quad (29)$$

where we now take $\pi_\gamma^*(a|s, g)$ as a point mass at $\mu(s, g, h_\gamma(s, g))$.

## F  ADDITIONAL EXPERIMENTS

In this section we study the performance of $C$-learning across different types of goals. We first evaluate $C$-learning for the Mini maze and Dubins' car environments, and partition their goal space into easy, medium and hard as shown in Figure 5. We run experiments for 3 (Mini maze) or 5 (Dubins' car) choices of random seed and take the average for each metric and goal stratification, plus/minus the standard deviation across runs.

The Mini maze results are shown in Table 2. We see that $C$-learning beats GCSL on success rate, although only marginally, with the bulk of the improvement observed on the reaching of hard goals. Both methods handily beat HER on success rate, and note that the path length results in the HER section are unreliable because very few runs reach the goal.

The Dubins' car results are shown in Table 3. We see that $C$-learning is the clear winner here across the board, with GCSL achieving a somewhat close success rate, and HER ending up with paths which are almost as efficient. Note that this result is the quantitative counterpart to the qualitative trajectory visualizations from Figure 4c. As mentioned in the main manuscript, we also tried explicitly enforcing monotonicity of our $C$-functions using the method of Sill (1998), but we obtained success rates below $50\%$ in Dubins' car when doing so.

We also compare $C$-learning to naive horizon-aware $Q$-learning, where instead of sampling $h$ as in $C$-learning, where consider it as part of the state space and thus sample it along the state in the replay buffer. Results are shown in the left panel of Figure 6. We can see that our sampling of $h$ achieves the same performance in roughly half the time. Additionally, we compare against sampling $h$ uniformly in the right panel of Figure 6, and observe similar results.

---

**Algorithm 2:** Training C-learning (TD3) Version

---

**Parameter:** $N_{\text{explore}}$: Number of random exploration episodes
**Parameter:** $N_{\text{GD}}$: Number of goal-directed episodes
**Parameter:** $N_{\text{train}}$: Number of batches to train on per goal-directed episode
**Parameter:** $N_{\text{copy}}$: Number of batches between target network updates
**Parameter:** $\alpha$: Learning rate

1   $\theta_1, \theta_2 \leftarrow$ Initial weights for $C^*_{\theta_1}, C^*_{\theta_2}$
2   $\theta'_1, \theta'_2 \leftarrow \theta_1, \theta_2$         `// Copy weights to target network`
3   $\phi \leftarrow$ Initial weights for $\mu_\phi$
4   $\mathcal{D} \leftarrow [\,]$        `// Initialize experience replay buffer`
5   $n_{\text{b}} \leftarrow 0$          `// Counter for training batches`
6   **repeat** $N_{\text{explore}}$ **times**
7      $\mathcal{E} \leftarrow \texttt{get\_rollout}(\pi_{\text{random}})$       `// Do a random rollout`
8      $\mathcal{D}.\texttt{append}(\mathcal{E})$      `// Save the rollout to the buffer`
9   **repeat** $N_{\text{GD}}$ **times**
10      $g \leftarrow \texttt{goal\_sample}(n_{\text{b}})$       `// Sample a goal`
11      $\mathcal{E} \leftarrow \texttt{get\_rollout}(\pi_{\text{behavior}})$       `// Try to reach the goal`
12      $\mathcal{D}.\texttt{append}(\mathcal{E})$       `// Save the rollout`
13      **repeat** $N_{\text{train}}$ **times**
14         $\mathcal{B} := \{s_i, a_i, s'_i, g_i, h_i\}_{i=1}^{|\mathcal{B}|} \leftarrow \texttt{sample\_batch}(\mathcal{D})$      `// Sample a batch`
15         $\{y_i\}_{i=1}^{|\mathcal{B}|} \leftarrow \texttt{get\_targets}(\mathcal{B}, \theta')$     `// Estimate RHS of equation 18`
16         $\hat{\mathcal{L}}_1 \leftarrow -\frac{1}{|\mathcal{B}|}\sum_{i=1}^{|\mathcal{B}|} y_i \log C^*_{\theta_1}(s_i, a_i, g_i, h_i) + (1 - y_i)\log\left(1 - C^*_{\theta_1}(s_i, a_i, g_i, h_i)\right)$
17         $\hat{\mathcal{L}}_2 \leftarrow -\frac{1}{|\mathcal{B}|}\sum_{i=1}^{|\mathcal{B}|} y_i \log C^*_{\theta_2}(s_i, a_i, g_i, h_i) + (1 - y_i)\log\left(1 - C^*_{\theta_2}(s_i, a_i, g_i, h_i)\right)$
18         $\theta_1 \leftarrow \theta_1 - \alpha\nabla_{\theta_1}\hat{\mathcal{L}}_1$       `// Update weights`
19         $\theta_2 \leftarrow \theta_2 - \alpha\nabla_{\theta}\hat{\mathcal{L}}_2$       `// Update weights`
20         **if** $n_{\text{b}} \mod \texttt{policy\_delay} = 0$ **then**
21            $\hat{\mathcal{L}}_{actor} \leftarrow -\frac{1}{|\mathcal{B}|}\sum_{i=1}^{|\mathcal{B}|} C^*_{\theta_1}(s_i, \mu_\phi(s_i, g_i, h_i), g_i, h_i)$
22            $\phi \leftarrow \phi - \alpha\nabla_\phi\hat{\mathcal{L}}_{actor}$       `// Update actor weight`
23            $\theta'_1 \leftarrow \theta_1 * (1 - \tau) + \theta'_1 * \tau$      `// Update target networks`
24            $\theta'_2 \leftarrow \theta_2 * (1 - \tau) + \theta'_2 * \tau$
25            $\phi' \leftarrow \phi * (1 - \tau) + \phi' * \tau$

26
27         $n_{\text{b}} \leftarrow n_{\text{b}} + 1$       `// Trained one batch`

---

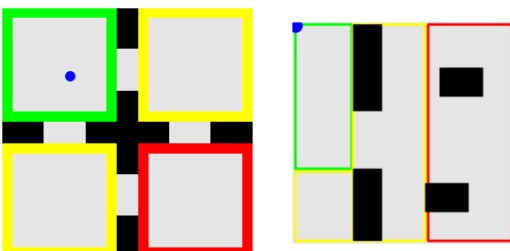

Figure 5: Partitioning of environments into easy (green), medium (yellow) and hard (red) goals to reach from a given starting state (blue). The left panel shows mini maze, and the right shows Dubins' car.

Table 2: Relevant metrics for the **Mini maze** environment, stratified by difficulty of goal (see Figure 5 (left) from appendix F). Runs in **bold** are either the best on the given metric in that environment, or have a mean score within the error bars of the best.

| METHOD | SUCCESS RATE | | | | PATH LENGTH | | | |
|---|---|---|---|---|---|---|---|---|
| | EASY | MED | HARD | ALL | EASY | MED | HARD | ALL |
| CAE | **99.44% ± 0.79%** | 97.52% ± 0.89% | **53.17% ± 25.77%** | **86.89% ± 6.56%** | 5.68 ± 0.02 | **14.84 ± 0.05** | 25.76 ± 0.61 | 13.81 ± 0.80 |
| GCSL | **99.17% ± 0.68%** | **99.72% ± 0.19%** | 37.74% ± 1.40% | 84.06% ± 0.61% | 5.65 ± 0.04 | **14.81 ± 0.02** | 23.46 ± 0.22 | 13.10 ± 0.02 |
| HER | 22.78% ± 15.73% | 21.63% ± 9.68% | 5.51% ± 7.79% | 17.87% ± 7.90% | 4.73 ± 0.63 | 13.94 ± 1.40 | 23.50 ± 0.00 | 11.58 ± 2.12 |

Table 3: Relevant metrics for the **Dubins' car** environment, stratified by difficulty of goal (see Figure 5 (right) from appendix F). Runs in **bold** are either the best on the given metric in that environment, or have a mean score within the error bars of the best.

| METHOD | SUCCESS RATE | | | | PATH LENGTH | | | |
|---|---|---|---|---|---|---|---|---|
| | EASY | MED | HARD | ALL | EASY | MED | HARD | ALL |
| CAE | **97.44% ± 2.23%** | **86.15% ± 4.06%** | **81.14% ± 4.02%** | **86.15% ± 2.44%** | **6.66 ± 0.91** | **16.16 ± 1.47** | **21.01 ± 1.50** | **16.45 ± 0.99** |
| GCSL | 93.85% ± 4.76% | 76.62% ± 7.11% | 75.68% ± 11.05% | 79.69% ± 6.35% | 17.58 ± 5.81 | 31.36 ± 3.99 | 41.85 ± 9.50 | 32.64 ± 6.16 |
| HER | 51.79% ± 12.60% | 55.38% ± 3.77% | 47.95% ± 10.20% | 51.25% ± 6.48% | 9.30 ± 2.37 | 18.06 ± 1.90 | 26.72 ± 2.84 | 20.13 ± 1.66 |

We also compare $C$-learning against TDMs Pong et al. (2018) for motion planning tasks. Results are in Figure 7. As mentioned in the main manuscript, TDMs assume a dense reward function, which $C$-learning does not have access to. In spite of this, $C$-learning significantly outperforms TDMs.

Finally, we also compare against the non-cumulative version of $C$-learning ($A$-learning), and the discount-based recursion ($D$-learning) in Dubins' car. For $D$-learning, we selected $\gamma$ uniformly at random in $[0, 1]$. Results are shown in table 4. We can see that indeed, $C$-learning outperforms both alternatives. Curiously, while underperformant, $A$-learning and $D$-learning seem to obtain shorter paths among successful trials.

## G    EXPERIMENTAL DETAILS

**C-learning for stochastic environments**: As mentioned in the main manuscript, we modify the replay buffer in order to avoid the bias incurred by HER sampling (Matthias et al., 2018) in non-deterministic environments. We sample goals independently from the chosen episode. We sample $g_i$ to be a potentially reachable goal from $s_i$ in $h_i$ steps. For example, if given access to a distance $d$ between states and goals such that the distance can be decreased at most by $1$ unit after a single step, we sample $g_i$ from the set of $g$'s such that $d(s_i, g) \leq h_i$. Moreover, when constructing $y_i$ (line 16 of algorithm 1), if we know for a fact that $g_i$ cannot be reached from $s_i'$ in $h_i - 1$ steps, we output $0$ instead of $\max_{a'} C^*_{\theta'}(s_i', a', g_i, h_i - 1)$. For example, $d(s_i', g_i) > h_i - 1$ allows us to set $y_i = 0$. We found this practice, combined with the sampling of $g_i$ described above, to significantly improve performance. While this requires some knowledge of the environment, for example a metric over states, this knowledge is often available in many environments. For frozen lake, we use the $L_1$ metric to determine if a goal is not reachable from a state, while ignoring the holes in the environment so as to not use too much information about the environment.

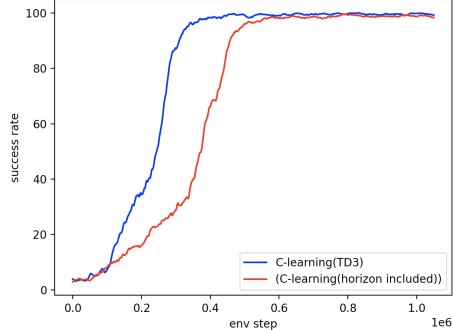
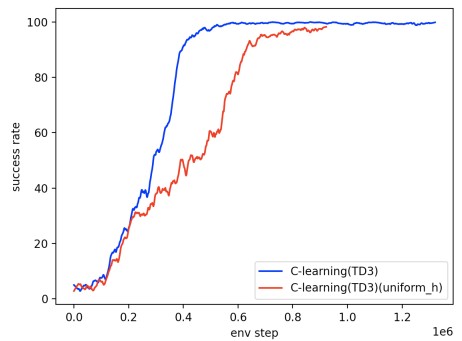

Figure 6: Success rate throughout training of $C$-learning (blue) and naive horizon-aware $Q$-learning (red) on FetchPickAndPlace-v1 (left); and $C$-learning (blue) and $C$-learning with uniform $h$ sampling (red) on FetchPickAndPlace-v1 (right).

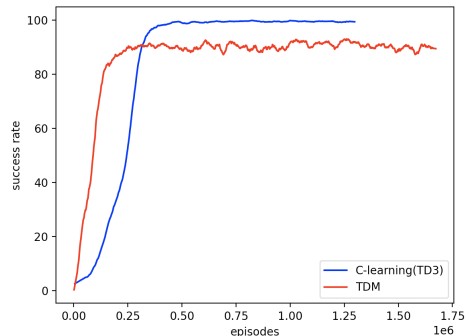 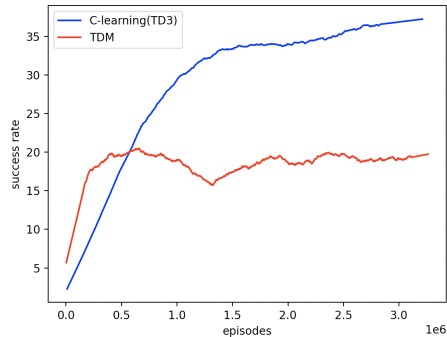

Figure 7: Success rate throughout training of $C$-learning (blue) and TDMs (red) in FetchPickAndPlace-v1 (left) and HandManipulatePenFull-v0 (right).

Table 4: Relevant metrics for the **Dubins' car** environment, stratified by difficulty of goal (see Figure 5 (right) from appendix F). Runs in **bold** are either the best on the given metric in that environment, or have a mean score within the error bars of the best.

| METHOD | SUCCESS RATE | | | | PATH LENGTH | | | |
|---|---|---|---|---|---|---|---|---|
| | EASY | MED | HARD | ALL | EASY | MED | HARD | ALL |
| $C$-learning | $\mathbf{97.44\% \pm 2.23\%}$ | $\mathbf{86.15\% \pm 4.06\%}$ | $\mathbf{81.14\% \pm 4.02\%}$ | $\mathbf{86.15\% \pm 2.44\%}$ | $6.66 \pm 0.91$ | $16.16 \pm 1.47$ | $21.01 \pm 1.50$ | $16.45 \pm 0.99$ |
| $A$-learning | $93.16\% \pm 3.20\%$ | $\mathbf{89.74\% \pm 4.41\%}$ | $62.88\% \pm 16.50\%$ | $78.12\% \pm 6.75\%$ | $\mathbf{5.84 \pm 0.59}$ | $13.29 \pm 0.28$ | $16.94 \pm 0.18$ | $12.83 \pm 0.39$ |
| $D$-learning | $86.32\% \pm 12.27\%$ | $34.36\% \pm 15.71\%$ | $2.27\% \pm 3.21\%$ | $30.21\% \pm 8.47\%$ | $\mathbf{5.62 \pm 0.25}$ | $\mathbf{11.44 \pm 0.26}$ | $\mathbf{13.97 \pm 0.00}$ | $\mathbf{7.91 \pm 0.82}$ |

### G.1 FROZEN LAKE

For all methods, we train for 300 episodes, each one of maximal length 50 steps, we use a learning rate $10^{-3}$, a batch size of size 256, and train for 64 gradient steps per episode. We use a 0.1-greedy for the behavior policy. We use a neural network with two hidden layers of respective sizes 60 and 40 with ReLU activations. We use 15 fully random exploration episodes before we start training. We take $p(s_0)$ as uniform among non-hole states during training, and set it as a point mass at $(1, 0)$ for testing. We set $p(g)$ as uniform among states during training, and we evaluate at every goal during testing. For $C$-learning, we use $\kappa = 3$, and copy the target network every 10 steps. We take the metric $d$ to be the $L_1$ norm, completely ignoring the holes so as to not use too much environment information. For the horizon-independent policy, we used $\mathcal{H} = \{1, 2, \ldots, 50\}$ and $\alpha = 0.9$. We do point out that, while $C$-learning did manage to recover the safe path for large $h$, it did not do always do so: we suspect that the policy of going directly up is more likely to be explored. However, we never observed GCSL taking the safe path.

### G.2 MINI MAZE

We again train for 3000 episodes, now of maximal length 200. We train 32 gradient steps per episode, and additionally decay the exploration noise of the $C$-learning behavior policy throughout training according to $\epsilon = 0.5/(1 + n_{GD}/1000)$. The network we used has two hidden layers of size 200 and 100 with ReLU activations, respectively. While the environment is deterministic, we use the same replay buffer as for frozen lake, and take the metric $d$ to be the $L_1$ norm, completely ignoring walls. We found this helped improve performance and allowed $C$-learning to still learn to reach far away goals. We also define $\mathcal{H}(s, g) := \{\|s - g\|_1, \|s - g\|_1 + 1, \ldots, h_{\max}\}$, with $h_{\max} := 50$, as the set of horizons over which we will check when rolling out the policies. We take $p(s_0)$ as a point mass both for training and testing. We use $p(g)$ during training as specified in Algorithm 1, and evaluate all the methods on a fixed set of goals. Additionally, we lower the learning rate by a factor of 10 after 2000 episodes. All the other hyperparameters are as in frozen lake. Figure 5 shows the split between easy, medium, and hard goals.

### G.3 DUBINS' CAR

We train for 4500 episodes, each one with a maximal length of 100 steps and using 80 gradient steps per episode. We use a neural network with two hidden layers of respective sizes 400 and 300 with ReLU activations. We take the metric $d$ to be the $L_\infty$ norm, completely ignoring walls, which we only use to decide whether or not we have reached the goal. We take $p(s_0)$ as a point mass both for

training and testing. We use $p(g)$ during training as specified in Algorithm 1, and evaluate all the methods on a fixed set of goals. All other hyperparameters are as in frozen lake. The partition of goals into easy, medium and hard is specified in Figure 5, where the agent always starts at the upper left corner.

### G.4 FETCHPICKANDPLACE-V1

We train with a learning rate of 0.0001 and batch size of 256. We take $64$ gradient steps per episode, and only update $\phi$ with half the frequency of $\theta$. We use $40000$ episodes for FetchPickAndPlace-v1. All the other hyperparameters are as in Dubins' car.

### G.5 HANDMANIPULATEPENFULL-V0

We train with a learning rate of 0.0001 and batch size of 256. We take $80$ gradient steps per episode, and only update $\phi$ with half the frequency of $\theta$. We use $60000$ episodes for HandManipulatePenFull-v0. All the other hyperparameters are as in Dubins' car.

