# OpenReview forum: "C-Learning: Horizon-Aware Cumulative Accessibility Estimation"
_ICLR.cc/2021/Conference — ICLR 2021 Poster_

### Official Review · AnonReviewer1 · 2020-10-25

**Rating:** 6
**Confidence:** 4

**Review:**

The paper highlights a problem in existing goal-reaching RL agents, in that they do not explicitly allow for trading off speed (how fast you reach the goal) and reliability (how often you reach the goal). While this tradeoff is implicitly determined by the discount factor in training, the paper asserts that in practice the ability to more flexibly determine this during inference is more desirable. Given this shortcoming of existing work, the paper then proposes "C-Learning" which learns a policy conditioned on both a goal and a desired horizon (h) -- i.e., a time limit on the policy. The paper presents favorable results of C-Learning on a few simulated domains compared to existing goal-reaching RL agents.

Strengths:

-- The proposed problem in standard goal-reaching agents is convincing.

-- I appreciate the level of detail in the algorithmic description along with the pseudocode. It was mostly easy to follow, except in a few small places (see weaknesses).

-- The experiments appear to have been performed carefully, with a helpful demonstration of the trade-off enabled by the proposed algorithm.

Weaknesses:

-- I found the small section about "Horizon Independent Policies" very confusing. For example, in the definition of M, since we know C* is increasing with h, does the max_h just reduce to setting h=infty. And if so, isn't the maximal value simply either 0 or 1 (measuring reachability)? And how is this max efficiently computed? More generally, why is this specific gamma-conditioned behavior policy necessary?

-- The various algorithmic details in 3.1 are worrying, as they suggest that a number of things beyond the basic C-Learning paradigm are necessary in practice. While I understand that this is unavoidable in any algorithm, some of these algorithmic details are exceedingly specific to the domain. For example, "C-function clipping" relies on knowing reachability in the environment, which is arguably as difficult as learning a goal-reaching policy in many cases!

-- The introduction mentions evaluating on domains from Nachum 2018, Peng 2018, and Zhang 2020. However, as far as I can tell, none of these domains are actually evaluated on? If you're not going to evaluate on these domains, I suggest removing this sentence from the paper.

-- While I am convinced of the problem the paper claims to solve, I am not convinced that C-learning is necessarily the best solution. I can imagine a number of other approaches which may perform better, worse, or about the same. For example, why not simply learn a policy pi(a|s, g, gamma) for all possible gamma in [0, 1)? Or alternatively, why not use one of the risk-sensitive policy learning approaches in the safe RL or constrained MDP literature (e.g., Lyapunov Safe RL)?

-- Learning a horizon-conditioned Q function was also proposed in TDM (https://arxiv.org/abs/1802.09081). How does C-learning relate and compare to this existing technique?

---

> ### Author Response · Authors · 2020-11-18
> **Rebuttal**
>
> Thank you for your review and constructive feedback.
>
> -Horizon independent policies: While the true C* is indeed non-decreasing in h, note that we take the maximum over a finite collection of potential horizons, denoted H in the text. While C* is always 0 or 1 for deterministic environments, this is not the case for stochastic environments, and thus the maximum of C* needs not be binary. Additionally, in practice we are using the learned C* function and not the true C* function. Small errors could cause e.g. the maximum over actions of C* to be 0.999, even in deterministic cases. Such a value should almost certainly be treated as 1 for the purpose of action selection, and converting to a horizon independent policy achieves this. Gamma (alpha in the updated version) can be thought of as a safety parameter in this context.
>
> In continuous environments we learn C* with the help of a critic, and we use the same critic to obtain the maximum over actions (see appendix C).
>
> Recovering horizon-unaware policies from horizon-aware ones is necessary for several reasons. For one, we used horizon-unaware policies with added randomness as the behavior policy when interacting with the environment for exploration. We also needed to compare against existing goal reaching methods, and the state-of-the-art techniques in goal reaching did not incorporate horizon information. For a fair comparison we needed to use tasks where a horizon is not specified, only a goal. We have updated the manuscript to convey these points in a clearer way.
>
> -Algorithmic details: As mentioned in other replies, after conducting additional experiments we have observed that C-clipping is not important to C-learning. We had used C-clipping in simpler domains where it was easy to determine reachability. The experiments we report on in the main text no longer use C-clipping.
>
> -References in introduction: We had mentioned several distinct domains where we believed  C-learning could be extended. We did not mean to imply that we were comparing our results to theirs. To avoid any misunderstanding we have removed these comments from the introduction as you suggested, and added them to a discussion in Section 6.
>
> -Alternative schemes: Thank you, including gamma as a conditioning variable is indeed another reasonable approach. We have now included this approach in the appendix, along with a detailed explanation of why using h is better. We also point out that while safety is an important research direction, when we say "reaching the goal reliably" we do not mean that there is a safety constraint during exploration. Rather we simply mean that, at test time, we can tune the agent's horizon to reach the goal reliably.
>
> -Thank you for pointing out TDMs. That work is indeed relevant and warrants discussion, which we have now included in the paper, but C-learning cannot be derived as a special case of TDMs. While TDMs use a similar recursion as ours, in practice they use an L1 distance reward, which significantly differs from our goal-reaching indicators. This is an important difference as TDMs operate under dense rewards, while we use sparse rewards, making the problem significantly more challenging. Additionally, distance between states in nonholonomic environments like Dubins' car or motion planning tasks is very poorly described by L1 distance; L1 distance is thus not a well-engineered reward in some of our environments. TDMs recover policies from their horizon-aware Q-function in a different manner to ours, not enabling the speed vs reliability trade-off. The recursion presented for TDMs is used by definition, whereas we have provided a mathematical derivation of the Bellman equation for C-functions along with proofs of additional structure. We also note that our motivation is different than that of TDMs, that we use a different loss (binary cross-entropy) which results in unbiased estimates of the objective's gradients (enabled as a consequence of C-functions being probabilities), and that we prove that optimal C-functions are monotonic. Finally, we also point out that TDMs sample horizons uniformly at random, which we will include in our ablation update.

---

> > ### Comment · AnonReviewer1 · 2020-11-23
> > **Thanks**
> >
> > Thanks for the clarifications and updates to the paper. I will update my score accordingly.
> >
> > Regarding "alternative schemes" -- it would be nice if there was a comprehensive empirical comparison, rather than just a rhetorical argument.

---

> > > ### Author Response · Authors · 2020-11-24
> > > **Discussion**
> > >
> > > Thank you for your suggestion.
> > > Due to limited time available today, we suspect the experiments will not finish in time.
> > >
> > > We will perform the experiments and include them on the C-learning website: [https://sites.google.com/view/learning-cae/](https://sites.google.com/view/learning-cae/).
> > > These will also be included in the manuscript before the camera-ready deadline to further include them on our paper.

---

### Official Review · AnonReviewer2 · 2020-10-28

**Rating:** 6
**Confidence:** 3

**Review:**

# Update after the rebuttal

Thank you for including the comparison to TDMs - the results are now much more convincing.
I think this should be one of the main results in the paper (which should be presented in the main text rather than in the appendix).
Also, I'd suggest revising the paper to highlight new contributions compared to TDMs in the intro/method section.
I increased the score assuming that the authors will reflect my final suggestions.

Regarding distributional RL and cross-entropy loss, distributional RL does allow a probability distribution as target (not just either 0 or 1 and just like yours). I'd suggest the authors to take a closer look at the paper and clarify differences (I still think the cross entropy loss is a special case of distributional RL with discount factor of 1).

# Summary
This paper proposes cumulative accessibility functions that estimate horizon-aware value function for goal-reaching RL problems. Specifically, the proposed C-function estimates the probability of reaching the goal state within a time budget (say $h$ steps). The paper derives Bellman-equation for C-function along with several properties. The experimental results on several continuous control domains show that the proposed method can balance between reliability and speed at test-time.

Pros
* The paper presents a new method that can balance between reliability and speed for goal-reaching RL problems.

Cons
* The novelty seems limited because the proposed method is quite similar to [1].
* Comparison to HER and GCSL is not fully convincing.

# Novelty
* My major concern is that this paper does not cite or discuss the most relevant prior work [1]. The proposed C-function seems like a special case of the TD model proposed by [1], which also learns Q(s, a, g, t) as a distance estimation ||s-g||. The TD model has almost the same Bellman equation as Equation (3) and it can also trade-off reliability and speed at test-time. It seems like the only difference is the definition of the reward function (distance v.s. 1/0). It is important for this paper to compare against [1] throughout the paper.
* The cross entropy loss (Equation 5) is not particularly novel, because it can be viewed as Distributional RL without discounting.

# Quality
* The empirical result shows the trade-off between reliability and speed nicely, which motivates the proposed method well.
* The comparison to HER and CGSL was mostly conducted on non-standard domains. The claim about faster learning would be more convincing if the experiment was conducted on more standard domains (e.g., MuJoCo, robotics manipulation)

# Clarity
* The motivation behind using a horizon-independent policy for behavior is not clear.
* The problem considered in this paper is not clearly defined. Is this self-supervised learning setting (where there is no final goal) or goal-reaching RL setting? The Algorithm 1 sounds like the former, while the experiment seems like the latter.

[1] Vitchyr Pong et al., Temporal Difference Models: Model-Free Deep RL for Model-Based Control.
[2] Marc G. Bellemare et al., A Distributional Perspective on Reinforcement Learning.

---

> ### Author Response · Authors · 2020-11-18
> **Rebuttal**
>
> Thank you for your review and constructive feedback.
>
> -Novelty:
>
> 1. Thank you for pointing out TDMs. That work is indeed relevant and warrants discussion, which we have now included in the paper, but C-learning cannot be derived as a special case of TDMs. While TDMs use a similar recursion as ours, in practice they use an L1 distance reward, which significantly differs from our goal-reaching indicators. This is an important difference as TDMs operate under dense rewards, while we use sparse rewards, making the problem significantly more challenging. Additionally, distance between states in nonholonomic environments like Dubins' car or motion planning tasks is very poorly described by L1 distance; L1 distance is thus not a well-engineered reward in some of our environments. TDMs recover policies from their horizon-aware Q-function in a different manner to ours, not enabling the speed vs reliability trade-off. The recursion presented for TDMs is used by definition, whereas we have provided a mathematical derivation of the Bellman equation for C-functions along with proofs of additional structure. We also note that our motivation is different than that of TDMs, that we use a different loss (binary cross-entropy) which results in unbiased estimates of the objective's gradients (enabled as a consequence of C-functions being probabilities), and that we prove that optimal C-functions are monotonic. Finally, we also point out that TDMs sample horizons uniformly at random, which we will include in our ablation update.
>
> 2. We are now also citing "A Distributional Perspective on Reinforcement Learning" as they do indeed use a cross-entropy loss. We do point out however that the settings are quite different, and that we use the binary cross entropy to avoid incurring in bias as is done when using the square loss (which incurs in bias by using samples to estimate the square of an expectation).
>
> -Quality: Please refer to our updated manuscript as we believe our new results are extremely convincing, doubling HER + TD3's performance on a difficult motion planning task, HandManipulatePenFull-v0, which is an unsolved OpenAI Robotics benchmark.
>
> -Clarity: Please see our updated manuscript, which now more clearly motivates our method. The problem we consider is multi-goal reaching where each episode will have a different goal, so this is not a self-supervised setting. However, as a consequence of learning to reach any goal, we can at test time fix any goal and try to reach it. The experiments that might at a first glance suggest otherwise (frozen lake and the plots for Dubins' car) were not obtained by fixing the goal throughout training. Instead, the agent learns to reach all possible goals, and we then fix the goal to produce the figures in which we show paths to the goal. The results in our table are indeed averaged over possible goals. We have clarified this in the manuscript.

---

> > ### Comment · AnonReviewer2 · 2020-11-24
> > **Thanks for the clarification but there are some remaining concerns.**
> >
> > **Regarding TDMs**
> >
> > Thanks for the clarification. I was wrong about the speed vs reliability trade-off (TDMs cannot achieve this). There are obvious differences between TDMs and C-learning as you described. However, my point was that in the end the differences mainly come from what kind of reward function is considered. TDMs considers a distance-based reward function, whereas C-learning considers a more sparse reward function that only gives reward of +1 when it reaches the goal. In the C-learning setting, the semantics of return naturally becomes reachability (probability of reaching goal), which leads to most of the new results in this paper (new Bellman equation, speed vs reliability trade-off, etc).
> >
> > As you said TDMs define the horizon-aware value function by defining the recursion followed by their explanation about what this horizon-aware value function means, whereas C-function is defined more formally, from which the recursion (Bellman equation) is derived. I do agree that C-learning is more nicely defined/derived than TDMs. However, my point was that the Bellman equation for C-learning is not entirely new (i.e., one could have defined TDMs formally and derived Bellman equation from it like C-learning).
> >
> > **Regarding cross entropy loss**
> >
> > I do not understand why your cross entropy loss is different from Distributional RL. Distributional RL maintains a distribution of possible returns (either 0 or 1 in your case) and apply Bellman equation from it. My intuition is that applying distributional RL loss with discount factor of 1 directly gives your cross entropy loss (both y and C are interpreted as return). Also, I do not follow why Q-learning is biased, whereas C-learning is unbiased. Is this about the overestimation issue caused by taking max over stochastic samples? Can you elaborate this further?
> >
> > **Overall**
> >
> > To be clear, I agree that this paper has several nice ideas (e.g., considering reachability reward function as opposed to distance-based reward function for goal-reaching RL problems, speed vs reliability trade-off, more formally defined horizon-aware value function + derivation of Bellman equation from it). However, the current version still gives me an impression that this paper was the first that introduces 1) horizon-aware value function + the new Bellman equation for horizon-aware value function until Page 6 where TDM is first mentioned, 2) and the cross entropy loss. Instead, it would be better to discuss what is new and novel compared to TDMs in a fair way throughout the paper (for example, extending the horizon-aware value function idea of TDMs to reachability reward function (which enables speed vs reliability tradeoff) seems like a key contribution). Also, an empirical comparison to TDM (distance-based reward function) would be important to justify the use of reachability formulation.

---

> > > ### Author Response · Authors · 2020-11-24
> > > **Additional rebuttal**
> > >
> > > Thank you for your reply.
> > >
> > > 1. Regarding TDMs: The sparse reward setting is more natural and generally considered harder. Hence C-Learning formalizes a harder problem and has strong empirical results as well. While one could build on TDMs to drive a Bellman recursion to the proposed C-Learning, it was neither referred to in TDM or otherwise. As the R2 mentioned, this is indeed a first formal presentation of such a result. We believe in hindsight this comparison and presentation does look obvious, however the algorithmic leap from TDM to C-Learning required multiple iterations of algorithm. Additionally, we have updated our manuscript and **now include a comparison against TDMs (appendix E, figure 7)**, where we significantly outperform them.
> > >
> > > 2. Using binary cross entropy is not distributional RL because the distribution corresponding to this loss is the Bernoulli distribution, and C-functions can have values outside of {0,1} (any value in the [0,1] interval is a valid value). Q-learning with a **square loss is biased as a result of estimating the square of an expectation by squaring a Monte Carlo estimate of the expectation**. To see this, consider the usual Q-learning objective:
> > > $$
> > > \mathbb{E}[\mathcal{L}(Q_\theta^*(s, g, a), y)]
> > > $$
> > > where the expectation is with respect to tuples (s, a, g) and the target y is itself an expectation with respect to s'|s, a. In practice the target cannot be evaluated exactly, and thus a single sample Monte Carlo estimate is used (by using s' from the replay buffer). This results in an unbiased estimate of y, but after applying the loss $\mathcal{L}$, results in a biased estimate of \mathcal{L}(Q_\theta^*(s, g, a), y), and thus a biased estimate of the objective as well. However, when $\mathcal{L}$ is linear in its second argument, as is the case with the binary cross entropy, the estimate of $\mathcal{L}(Q_\theta^*(s, g, a), y)$ remains unbiased.
> > >
> > > 3. Overall: **We will cite and add text about TDMs early in manuscript, and further clarify the bias discussion**. As suggested the major differences are in nontrivial extensions of the horizon-aware value function idea of TDMs to reachability reward function (which enables speed vs reliability tradeoff) and the cross-entropy loss. Again, we point out that we now have an empirical comparison against TDMs (appendix E, figure 7).

---

### Official Review · AnonReviewer4 · 2020-10-29
**This submission calls for a more careful analysis of the proposed method components**

**Rating:** 6
**Confidence:** 4

**Review:**

### Summary
The paper proposes learning horizon-aware goal conditioned policies by integrating horizon dependency in value-based methods. The new formalism defines the value function (cumulative accessibility function) as the probability of reaching a certain goal from a given state within a fixed horizon. This allows trading-off speed and reliability.

### Main contributions
- The C-learning is a different approach to integrate temporal abstractions in value-based method.
- The derivation of the cumulative accessibility function in analogy to the standard Q-learning is principled and very interesting.
- The proposed approach is compatible with the algorithmic adaptations of Q-learning to continuous action spaces.

### Main concerns (and comments)
- The exploration problem has been ignored in this work while it is quite central. The standard Q-learning (if in the undiscounted return like your case) would go for the higher expected return which would correspond to the policy of the highest success probability, when provided with a satisfying exploration. In addition to this, alternative standard methods are horizon-penalized (e.g. through the discount factor). These point should be considered in this study and in the comparison to other methods.
- It is claimed that encoding monotonicity in h in the architecture hurt performance. A hypothesis is provided by not tested. More importantly, did the monotonicity emerge naturally from training ?
- Regarding C-function clipping. How is "if we know for a fact that g_i cannot be reached ..." implemented practically ? It is acknowledged that this requires "some knowledge of the environment, for example a metric over states". However, such a metric even when available does not always say how many steps separate the states (i.e continuous spaces).
- "[...] but also shown its monotonicity, a crucial component which facilitates neural network learning in the higher-dimensional goal-horizon space" : can you elaborate on this ? Is this claim referring to some property or known result on "neural network learning" ? If so any references ?
- "Surprisingly, GCSL learns to take the high-risk path, despite Ghosh et al.’s intention to incentivize paths that are guaranteed to reach the goal": Does your method uses a goal sampling strategy that provides more options to the agent (like a richer exploration would do) ? In other words, how much of the performance depends on the goal sampling strategy (ablation study) ?
- In Dubin's car, the set of reachable goals from a given state within some horizon does not have a trivial structure, how was the goal sampling implemented here ?
- Regarding the C-function clipping, can we imagine a goal conditioned Q-learning baseline that would be trained with a similar clipping, i.e how much of the performance is explained by this inductive bias (the C-clipping) ?
- "This is also impressive, as we are learning a more complicated objective, which contains information not recoverable by the alternatives." : This comment is misleading. Even if the objective is more complicated, the proposed method learns a more structured function and benefits (for free) from valuable information (like the C-function Clipping) while "alternatives" have to implicitly learn such information.
- "We did train for longer and observed decreased performance" : Does this mean the performance decreases for all the curves that were not plotted till convergence ? How do you explain this ? Also, Figure 7 shows averages without standard deviations.
- "We also note that, strangely, while C-learning dominates easy and medium goals in mini maze, it struggles to reach hard goals. We found this particularly puzzling given the good performance in the more challenging environments." : This deserves more attention and might suggest to consider the exploration problem (mentioned in the first comment) more seriously.

### Minor comment
- The objective (eq. 5) is quite confusing. Should it be minimized or maximized ?
- How does this compare to planning with options over different horizons ?

---

> ### Author Response · Authors · 2020-11-18
> **Rebuttal**
>
> Thank you for your review and constructive feedback.
>
> 1. We have updated our manuscript and now explain why horizons h should be preferred over discount factors as a way to obtain "horizon-penalized" methods (see section 3.1).
>
> 2. Note in the Dubins' car figure (4b) that the learned C* function is indeed monotonic without this constraint being imposed, as can be seen by the reachable sets increasing in size. We will include a comparison where monotonicity is explicitly enforced by the neural network architecture in the forthcoming ablation study. It is useful to contrast C-learning to the non-cumulative version (probability to reach the goal in exactly h steps) which we covered in appendix B. The main difference between these approaches is monotonicity. Our choice to focus on C-learning was based on early experiments where it performed better.
>
> 3. As mentioned in the general rebuttal, upon performing additional experiments, we observed that C-clipping is not required for effective learning. The experiments in the main text do not use any clipping anymore.
>
> 4. Our comment on monotonicity was based on our preliminary experiments where we compared C-learning to the non-cumulative version and observed better performance for C-learning. We have clarified this in the manuscript.
>
> 5. We use the same goal-sampling scheme for GCSL as in C-learning so C-learning is not being unfairly helped by additional exploration.
>
> 6. In our updated version, we are selecting goals uniformly during training and evaluation across all environments.
>
> 7. As mentioned above, in our updated version we are not zeroing out C*, and are not providing C-learning additional information about the environments.
>
> 8. The optimal C*-function may be more structured and contain more information, but we are showing that it can still be learned in a number of environment steps comparable to other methods. This is still possible without added information from clipping.
>
> 9. We have tidied up the figures in our updated manuscript to have consistent ranges which should address your concerns.
>
> 10. We have updated the experimental results for the simpler environments in Appendix E. C-learning takes longer paths to goals in these environments than GCSL, but is competitive on final distance.
>
> 11. The binary cross entropy is minimized. We have added a previously missing minus sign.
>
> 12. The relationship to options is relevant and very interesting. We have included a detailed discussion in section 6 of the manuscript, where we point out that, despite the parallels, C-learning does not fit within the options framework, and nothing in the options framework encourages a safety/reliability trade-off, nor is reachability information recovered in any way.

---

> > ### Comment · AnonReviewer4 · 2020-11-24
> > **Thank you for your the clarifications**
> >
> > Thank you for the clarifications and the revision.
> >
> > It is indeed interesting that the C-clipping is proved to not be necessary. I would suggest to still provide (as a complementary empirical analysis) a comparison to a clipped version as a measure of how successful the updated version is in recovering that information.
> >
> > Most of my concerns being addressed, I'll update my score accordingly.

---

> > > ### Author Response · Authors · 2020-11-24
> > > **Discussion**
> > >
> > > Thank you for your suggestion, due to limited time available today, we suspect the experiments will not finish in time.
> > > We will perform the experiments and include them on the C-learning website: https://sites.google.com/view/learning-cae/.
> > > These will also be included in the manuscript before the camera-ready deadline to further include them on our paper.

---

### Official Review · AnonReviewer3 · 2020-11-01
**Why no comparison with time horizon conditioned Q-learning?**

**Rating:** 5
**Confidence:** 3

**Review:**

The paper proposes C-learning, which is an essentially a horizon aware Q-learning. In a nutshell, the authors proposes changing the Q function from Q(s, a) to C(s, a, h) where s is the state, a is the action and h is allowed time horizon i.e. the agent should get to the goal state using less than h states. Since C can be framed as a modified Q, the authors demonstrate that familiar Q-learning properties such as Bellman property and backprop can be used for C-learning.

The paper is generally well-written and is easy to understand. The authors evaluated the performance of the proposed method on multiple hand crafted tasks which allow for multiple way of reaching to the goal with a varying hard time limit. The authors compared their method with GCSL (Ghosh et al., 2019) and HER (Andrychowicz et al., 2017) demonstrating an improved performance. The included website includes extra evaluations on a few robotic tasks.

The paper is suffering from multiple shortcomings:
1) Motivation. Although I agree that the proposed problem is important, the authors could provide more motivation. The listed examples demonstrate the problem however there is no real life example that visualizes the *importance* of the problem.

2) Comparison with existing goal reaching methods. As mentioned by the authors, the reachability problem can be formulated as in an existing RL framework and therefore the proposed method can also be compared with existing RL methods. The comparison section tries to do this but unfortunately it falters. I believe the comparison section should include existing methods with modified RL formulation so they are horizon aware (e.g. by including the horizon limit as input and training them by randomly selecting it). Doing this will highlight *why* C-learning is required and a simple Q-learning method cannot solve the problem at hand. There are other issues in Section 4 as well. For example Fig 7 has incomplete curves, the metrics are not clearly defined and it is not clear how the baselines have been evaluated.

3) Task selection. The paper evaluated the performance of their model on toy environments which is always a good idea. That enables the authors to clearly differentiate their proposed method from previous work (which is missing as I mentioned in 2). But, at the end of the day, it is important to visualize that the proposed method works on real challenges which is missing. I should mention that the website includes more realistic tasks and addresses this problem to some sense but the paper in its current form lacks.

Overall, I think the paper can use clearer motivation, comparison and evaluation.

---

> ### Author Response · Authors · 2020-11-18
> **Rebuttal**
>
> Thank you for your review and constructive feedback.
>
> 1. Motivation: We have modified the manuscript, and now more clearly motivate our work not just through the speed/reliability trade-off, but by arguing that C-learning is more sample-efficient than Q-learning for goal reaching because the C*-function is more tolerant to errors than the Q*-function.
>
> 2. Experiments: We have updated our manuscript and now include comparisons in currently unsolved environments. We note that we are comparing against GCSL and TD3 with HER, which are extremely competitive baselines for goal reaching. We will include an ablation study in the manuscript in the upcoming days showing how C-learning outperforms horizon-aware Q-learning where h is included as part of the state. Horizon aware Q-learning requires each state to be sampled with the horizon saved in the replay buffer. C-learning is more flexible, allowing experiences to be relabeled with different goals and horizons, making training more sample efficient.
>
> 3. Task selection: As mentioned above, additional results on much harder RL environments are now included in the updated manuscript.

---

> > ### Comment · AnonReviewer3 · 2020-11-24
> > **Re: Rebuttal**
> >
> > Thank you for your feedback. Given the update text and new experiments, I increased my score. However, I believe comparison with horizon aware Q-learning is still a key missing experiment which holds the paper back. I read the newly added sub section 3.1 and I find it convincing but didn't see any head to head experiment. Please let me know if you already added this experiment but I missed it.

---

> > > ### Author Response · Authors · 2020-11-24
> > > **Discussion**
> > >
> > > Thank you for your reply. We have indeed included a comparison against horizon-aware Q-learning on FetchPickAndPlace-v1 in appendix E (figure 6, left panel), where C-learning requires about half the interactions to achieve the same performance.

---

### Author Response · Authors · 2020-11-18
**General rebuttal**

We thank all the reviewers for their time and feedback. We have thoroughly updated our manuscript to address all the important points raised by the reviewers, including motivation, additional experiments in complex environments where we significantly outperform existing baselines, and a detailed discussion about Temporal Difference Models (TDMs). As an important note, upon performing ablation experiments we observed that C-clipping is not required for good empirical performance; this should address some reviewer's concerns about C-learning requiring too much information about the environment. We will make another update in the upcoming days including ablation studies in the appendix.

---

> ### Author Response · Authors · 2020-11-24
> **Manuscript update**
>
> As mentioned in our rebuttal, we have now included ablations and comparisons against TDMs in an updated version of our manuscript (see appendix E), and we indeed observe either better success rate, or much faster convergence from C-learning than from alternatives.

---

### Decision · Program_Chairs · 2021-01-07
**Final Decision**

**Decision:**

Accept (Poster)

**Comment:**

This paper introduces C-learning, an approach to integrate temporal abstractions to value-based methods. Specifically, it uses accessibility functions that estimate horizon-aware value functions for goal-reaching RL problems. Such an approach allows trading-off reliability and speed. After careful consideration I’m recommending the acceptance of this paper. The main weaknesses raised by the reviewers were addressed during the rebuttal, including the improvement of presentation and the introduction of new experiments and baselines. There were not many actionable criticisms left after the discussion and the reviewers acknowledged that the paper has improved since its first version.

For the final version of the manuscript, I recommend the authors to further take R2’s comments/suggestions into consideration. Further incorporating the discussion about TDMs in the main text will improve clarity, better position the paper, and increase its likelihood of having impact.